# Ultrathin two-dimensional porous organic nanosheets with molecular rotors for chemical sensing

Jinqiao Dong[1], Kang Zhang [1], Xu Li[1], Yuhong Qian[1], Hai Zhu[2], Daqiang Yuan [3], Qing-Hua Xu[2], Jianwen Jiang[1] & Dan Zhao [1]

Molecular rotors have played an important role in recent materials chemistry. Although several studies on functional materials containing molecular rotors have been reported for fluorescence sensing, this concept has yet to be realized in two-dimensional (2D) materials. Here we report the preparation of all-carbon, π-conjugated 2D porous organic nanosheets, named NUS-24, which contain flexible tetraphenylethylene (TPE) units as the molecular rotors. NUS-24 nanosheets exhibit high stability, large lateral size, and ultrathin thickness (2–5 nm). The dynamic TPE rotors exposed on the surface of NUS-24 nanosheets can be restricted in the aggregated state with different water fractions, which is reminiscent of the aggregation-induced emission mechanism, thereby leading to the size-selective turn-on fluorescence by volatile organic compounds. Significantly, the ultrathin 2D nanosheets and its composite membranes show much higher sensitivity and selectivity toward $Fe^{3+}$ ions and nitro-containing compounds sensing, suggesting their potential applications in explosive detection and environmental monitoring.

[1] Department of Chemical & Biomolecular Engineering, National University of Singapore, Singapore 117585, Singapore. [2] Department of Chemistry, National University of Singapore, Singapore 117543, Singapore. [3] State Key Laboratory of Structural Chemistry, Fujian Institute of Research on the Structure of Matter, Chinese Academy of Sciences, Fujian, Fuzhou 350002, China. Correspondence and requests for materials should be addressed to D.Z. (email: chezhao@nus.edu.sg)

The rational design of artificial molecular machines is an important objective in synthetic chemistry and functional materials research, and it has been achieved in the pioneering work by Stoddart[1–3], Feringa[4,5], and Leigh[6–8]. Among these molecular machines, fluorescent molecular rotors have received wide attention recently due to their importance in

broadening the fundamental understanding of dynamic molecular systems and their wide applications in areas including biological detection[9–13], materials chemistry[14,15], membrane engineering[16], and so on. Generally, the working mechanism is based on the different degree of restriction of the molecular rotors. For example, Garcia-Garibay and co-workers have

**Fig. 1** Design and synthesis of NUS-24 nanosheets. **a** Synthetic route of NUS-24 bulk powder by Sonogashira–Hagihara coupling reactions. **b** The schematic diagram for the exfoliation of NUS-24 from bulk layered material to 2D nanosheets. The phenyl rings with pink color represent dynamic TPE rotors whose motion can be liberated after exfoliation. **c** Theoretical calculations of energy barrier of TPE rotors in the constructed models

**Fig. 2** Spectroscopic characterization of NUS-24 nanosheets. **a** XPS spectra of TPE-1, bulk powder, and nanosheets of NUS-24. **b** Raman spectra of bulk powder and nanosheets of NUS-24. **c** UV–Vis spectra of TPE linkers, bulk powder, and nanosheets of NUS-24 in acetone solution. **d** Optical band gaps ($E_g$) of bulk powder and nanosheets of NUS-24 in solid state by diffuse reflectance. **e** Fluorescence emission spectra of TPE linkers (solid state), bulk powder, and nanosheets of NUS-24 in acetone solution (inset: fluorescence photographs of nanosheets solution with different concentrations in acetone. $\lambda_{ex}$ = 365 nm). **f** CIE coordinates of emission color of TPE-1, TPE-2, bulk powder, and nanosheets of NUS-24. **g** The frontier molecular orbital distributions of NUS-24 fragment by DFT calculations

successfully demonstrated a series of crystalline dynamic molecular rotor-based supramolecular assemblies[17], macrocycles[18], one-dimensional (1D) helical domains[19], and three-dimensional (3D) metal-organic frameworks (MOFs)[20]. Aprahamian and co-workers have recently reported the controlled fluorescence emission by restricting boron difluorohydrazone (BODIHY) rotors through viscosity adjustment[21]. Our group has also realized such controlled fluorescence emission through the restriction of the rotors using volatile organic compounds (VOCs) in 3D porous organic frameworks (POFs)[22] and 3D MOFs[23] for chemical sensing.

The last decade has witnessed the booming development of two-dimensional (2D) materials such as graphene and transition metal dichalcogenides (TMDs)[24,25]. Since the initial discovery of graphene[26], one of the key challenges in 2D materials is to go beyond graphene using synthetic chemistry. In this respect, Müllen and co-workers[27–29] have done pioneering work in nanographene chemistry. Very recently, Feng and co-workers synthesized a series of extended polycyclic aromatic hydrocarbons (PAHs) or graphene molecules for electronic device[30–33]. These novel 2D materials have large aspect ratios with fully exposed basal planes, and demonstrate good catalytic and

electrochemical properties as well as being promising platforms for chemical sensing and biosensing applications[34]. In particular, Zhang and co-workers demonstrated that ultrathin 2D MOF nanosheets[35] and 2D ternary chalcogenide nanosheets[36] can be used as fluorescence DNA sensors. Banerjee and co-workers also obtained covalent organic nanosheets to detect and distinguish VOCs[37]. Qian et al. fabricated 2D MOF nanosheets for metal ion sensing[38]. Given the above progress, we envision that the introduction of molecular rotors into 2D materials may be able to generate highly sensitive fluorescent chemical sensors due to the large surface area of 2D materials. In addition, the molecular rotors would enhance the sensitivity and selectivity of these fluorescent sensors for specific sensing applications.

Herein, we report the design and synthesis of a porous organic nanosheets (PONs), pure organic polymers with 2D layered structures, and nanosheet-like morphology, named NUS-24 containing flexible tetraphenylethylene (TPE) units as the molecular rotors. This can be used to circumvent the problem of instability in 2D MOF nanosheets. NUS-24 nanosheets with thicknesses of around 2–5 nm (3–6 layers) can be obtained via top-down solvent exfoliation from layered NUS-24 bulk powder. Interestingly, the presence of dangling TPE rotors exposed on the surface of NUS-24 nanosheets is evidenced by the varied responses to different water fraction and size-selective VOC sensing. We also apply NUS-24 nanosheets to chemical sensing of metal ions. In addition, composite membranes containing NUS-24 nanosheets are also fabricated and tested for practical sensing applications. Last but not least, density functional theory (DFT) calculations are conducted to estimate the highest energy occupied molecular orbital (HOMO)–lowest unoccupied molecular orbital (LUMO) energy profiles and provide insights into the experimentally observed sensing behavior.

## Results

**Synthesis and characterization of NUS-24 nanosheets**. As seen in Fig. 1, 1,2-bis(4-bromophenyl)-1,2-diphenylethene (TPE-1, Supplementary Figs. 1, 2) and 1,1,2,2-tetrakis(4-ethynylphenyl) ethane (TPE-2, Supplementary Figs. 1, 2) were employed to synthesize NUS-24 due to their rigid molecular conformation and strong directional effect that can easily generate the expected 2D layered structures[39]. In addition, the central olefin stators of TPE-1 linker are surrounded by two free peripheral phenyl rings, which can act as molecular rotors for turn-on fluorescent sensing, thereby forming a planar surface with protruding free dynamic TPE rotors on top. To microscopically elucidate the dynamic feature of the TPE rotors, we constructed 37 cluster models containing one TPE-1 linker and two TPE-2 linkers for evaluating rotational energy barrier (Supplementary Fig. 3; Supplementary Note 1). Figure 1c illustrates the potential energy as a function of rotor angle. Several minima are observed at 50°, 130°, 240°, and 320°, implying the existence of stable conformations. While at 90°, 180°, and 290°, maxima are seen with relatively higher energy for less favorable conformations. Three rotation barriers exist with the highest barrier of 14.0 kcal mol$^{-1}$ at 180°; and the other two barriers are 5.1 and 8.0 kcal mol$^{-1}$ at 50° and 290°, respectively. Thus, we believe that the TPE-1 linker contains dynamic molecular rotors suitable for the construction of highly efficient 2D nanomaterials with potential applications in molecular recognition and sensing.

As illustrated in Fig. 1a, layered bulk NUS-24 was successfully synthesized through Sonogashira–Hagihara coupling reaction with a yield of 84%, leading to an inherently layered porous framework with built-in TPE rotors. Compared to the poly-condensation reactions used in generating 2D covalent organic frameworks (COFs) with strong interlayer interactions[40], the C–C

coupling reaction used in NUS-24 forms strong covalent bonds within each layer, but only affords weak interlayer interactions. This is due to the irreversibility of C–C coupling reaction and the presence of twisted TPE rotors that may prevent effective interlayer packing. The resultant structure is a highly distorted aromatic framework that favors its exfoliation and dispersion into few layered 2D nanosheets[41,42]. Thereby, the free-standing ultrathin NUS-24 nanosheets can be readily obtained by liquid exfoliation of its bulk powder (Fig. 1b). In this process, solvent molecules can enter the interlamination spaces and expand them, and finally cleave the 2D layered bulk material into few layered nanosheets. We have tested a wide range of solvents with different surface tensions, and found that acetone, acetonitrile, and ethanol are good candidates for the exfoliation of NUS-24 bulk powder. As a result, the TPE rotors, which are originally restricted by the interlayer interactions between the adjacent packing layers in the bulk powder, can now be fully liberated and exposed on the surface of NUS-24 ultrathin nanosheets, contributing to enhanced sensitivity in chemical sensing applications (vide infra).

Fourier transform infrared spectroscopy (FT-IR) spectra show almost complete disappearance of C-Br vibration bands (around 590 cm$^{-1}$) in both NUS-24 bulk powder and NUS-24 nanosheets, indicating the completion of the cross-coupling reaction (Supplementary Fig. 4). X-ray photoelectron spectroscopy (XPS) spectra further confirm the vanishing of the Br1s peak (at around 70.7 eV)[22] in both bulk powder and nanosheets of NUS-24 (Fig. 2a). In addition, no obvious shift of the C1s binding energy is observed between bulk powder and nanosheets, suggesting identical chemical structures between these two. The chemical structure of bulk NUS-24 was further characterized by solid-state $^{13}$C CP/MAS NMR spectroscopy (Supplementary Fig. 5). The spectrum displays one main peak at 63.50 ppm assignable to TPE-2 linker. Raman spectra of bulk powder and as-exfoliated nanosheets show nearly identical main peaks located at around 1585 cm$^{-1}$ (Fig. 2b), which can be assigned to the carbon vibration of $sp^2$ hybridization[43]. This further confirms the structural similarity between bulk powder and exfoliated nanosheets. However, compared to NUS-24 bulk, the Raman spectrum of NUS-24 nanosheets shows a slight shift toward high wavenumber, which can be attributed to the ultrathin thickness of the 2D nanosheets[44]. Meanwhile, the ultraviolet–visible (UV–Vis) spectrum of the NUS-24 nanosheets has a blue shift of about 65 nm compared to that of the bulk powder (430 versus 365 nm), which is mainly due to the reduced interlayer π–π interaction in nanosheets (Fig. 2c; Supplementary Fig. 6). Such change of π–π interaction results in a sharp increase of optical band gap ($E_g$)[45], from 2.39 eV in bulk powder to 2.77 eV in nanosheets (Fig. 2d). In addition, the fluorescence emission of nanosheets has also blue-shifted by about 15 nm compared to that of bulk powder (535 versus 520 nm, Fig. 2e). The observed emission colors of TPE-1, TPE-2, NUS-24 bulk, and NUS-24 nanosheets are differentiable on the Commission Internationale de L'Eclairage (CIE) chromaticity diagram (Fig. 2f; Supplementary Fig. 7). We further performed DFT calculations to examine the electronic structures of NUS-24 (Fig. 2g). The calculated frontier orbitals of NUS-24 fragment indicate that the characteristics of their individual components are maintained due to the absence of electron-withdrawing group[46], which is helpful for fluorescence emission. For the nanosheets, the main UV–Vis absorption band centers at around 365 nm, corresponding to the HOMO to LUMO transition based on the DFT calculation. Moreover, the high electrostatic surface potential of NUS-24 fragment due to TPE rotors, obtained by DFT calculations (Supplementary Fig. 8), allows interaction with guest molecules for chemical sensing. Thermogravimetric analyses (TGA) show that both NUS-24 bulk powder and nanosheets are thermally stable up to 400 °C

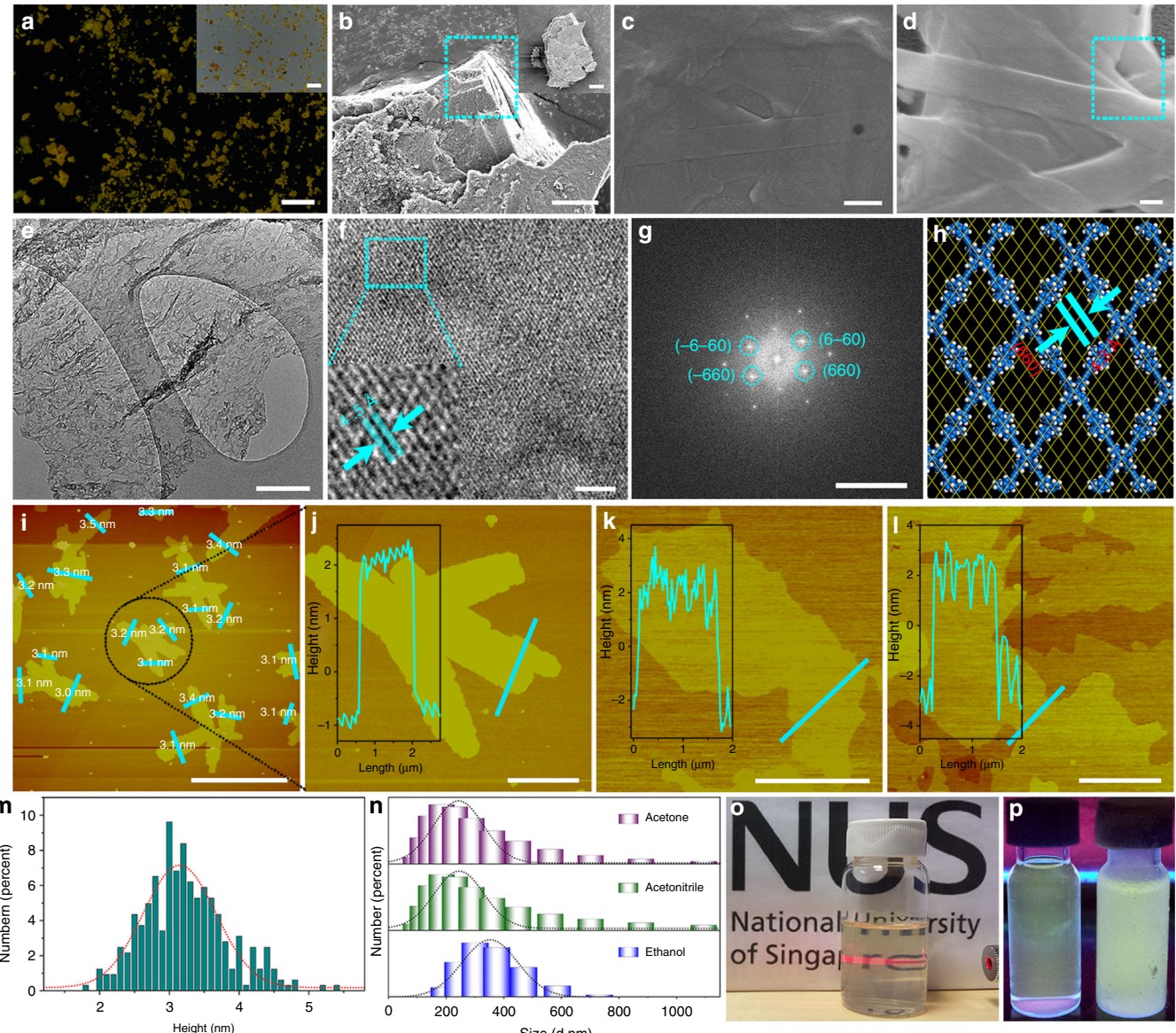

**Fig. 3** Morphology characterization of NUS-24 nanosheets. **a** Fluorescence photograph of NUS-24 bulk powder (scale bar, 3 mm. $\lambda_{ex} = 365$ nm). Inset of **a** corresponding optical photograph (scale bar, 3 mm). **b** FE-SEM image of NUS-24 bulk powder (scale bar, 10 μm. scale bar of inset of **b**, 30 μm). **c, d** FE-SEM images of NUS-24 nanosheets deposited on AAO substrate (scale bar, 500 nm in **c** and 200 nm in **d**). The circulated rectangular areas indicate the 2D layered structure of NUS-24. **e** TEM image of NUS-24 nanosheets (scale bar, 200 nm). **f** HR-TEM image of NUS-24 nanosheets featuring the planar lattice structure (scale bar, 5 nm. Inset: the lattice distance). **g** The fast Fourier transformation of **f** (scale bar, 5 1 per nm). **h** The lattice structure from (660) plane of the optimized NUS-24 crystal structure. **i–l** AFM images of NUS-24 nanosheets dispersed in acetone (**i, j**), acetonitrile (**k**), or ethanol (**l**) (scale bar, 10 μm in **i** and 2 μm in **j–l**. Inset: the height of AFM image for selective area). **m** The statistical height distribution of NUS-24 nanosheets based on 322 AFM measurements. **n** The DLS results of NUS-24 nanosheets in different solutions. **o** The optical photograph of NUS-24 nanosheets in acetone solution indicating strong Tyndall effect. **p** The fluorescent photograph of NUS-24 nanosheets in acetone solution at 298 K (left, solution state) and 77 K (right, solidified state) ($\lambda_{ex} = 365$ nm)

(Supplementary Fig. 9). Excellent chemical stability of bulk NUS-24 was also proven through soaking tests using water, hydrochloric acid (6 M), sulfuric acid (6 M), sodium hydroxide (8 M), and common organic solvents (Supplementary Table 5). The extremely high thermal and chemical stabilities make NUS-24 attractive for applications even under corrosive conditions.

Permanent porosity of bulk NUS-24 is demonstrated by its $N_2$ sorption isotherm at 77 K (Supplementary Fig. 10), which exhibits a type I sorption behavior with a Brunauer–Emmett–Teller (BET) surface area of 335 m² g⁻¹ and a total pore volume of 0.378 cm³ g⁻¹. Pore size distribution calculated using nonlocal density functional theory (NLDFT) reveals the presence of both

micropores (around 14 Å) and mesopores (around 28 Å). However, the BET surface area of NUS-24 nanosheets decreases after exfoliation in different organic solvent such as acetone (100 m² g⁻¹), ethanol (162 m² g⁻¹), and acetonitrile (92 m² g⁻¹). The total pore volume of nanosheets is also smaller than that of its bulk powder (0.13, 0.20, and 0.13 cm³ g⁻¹ after exfoliation in acetone, ethanol, and acetonitrile, respectively). Pore size distribution shows that the microporous characteristics almost disappear in the 2D nanosheets. This might be due to the disruption of the π−π stacking among layers by as-exfoliated nanosheets[47]. In addition, no diffraction peak can be observed in the powder X-ray diffraction (PXRD) pattern of both NUS-24

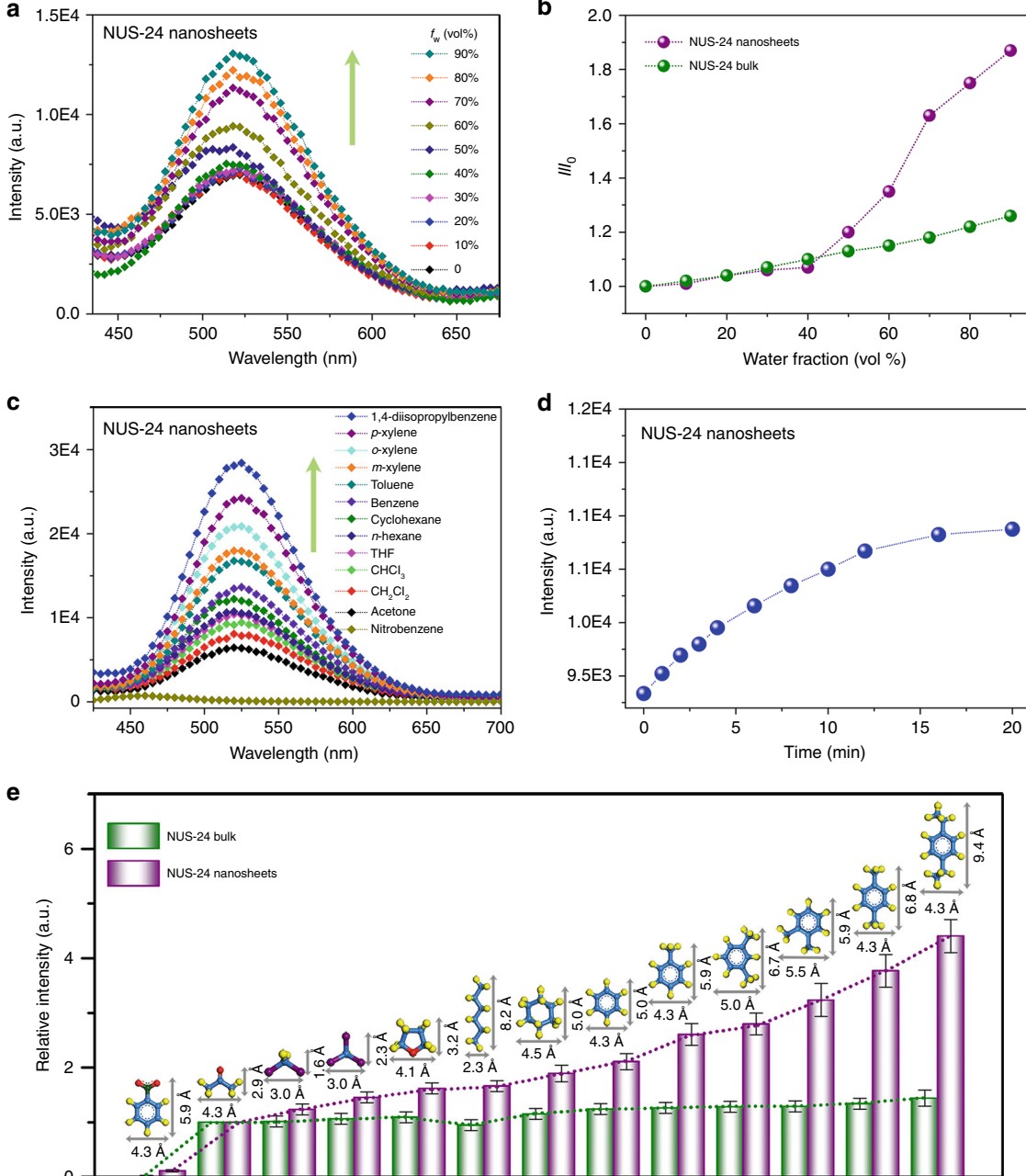

**Fig. 4** The dynamic behavior study of TPE rotors in NUS-24 nanosheets. **a** Fluorescent spectra of NUS-24 nanosheets ($c = 20\,\mu g\,mL^{-1}$, $\lambda_{ex} = 365\,nm$) in acetone and acetone/water mixtures. **b** Plot of relative emission intensity versus water fraction in acetone/water mixtures of NUS-24 nanosheets and bulk powder. **c** Fluorescence emission spectra of NUS-24 nanosheets in various VOC solutions at room temperature ($c = 20\,\mu g\,mL^{-1}$, $\lambda_{ex} = 365\,nm$). **d** Time-dependent fluorescence enhancement of NUS-24 nanosheets by adding 1,4-diisopropylbenzene (20% v/v in acetone). **e** Relative fluorescence intensity of bulk powder and nanosheets of NUS-24 in various VOC solutions ($I_R = I/I_{acetone}$). The error bars show average and range of measured values based on at least five repeats in every single measurement

bulk powder and its 2D nanosheets (Supplementary Fig. 11), indicating its amorphous nature similar to other polymers obtained via C–C cross-coupling reactions[22,48].

The expected 2D layered structure of NUS-24 bulk powder was confirmed by field-emission scanning electron microscopy (FE-SEM, Fig. 3a, b; Supplementary Fig. 12a, b). The sheet-like morphology of as-exfoliated NUS-24 supported on porous anodic aluminum oxide (AAO) substrate was also observed using FE-SEM (Fig. 3c, d; Supplementary Fig. 12c, d). High-resolution transmission electron microscopy (HR-TEM) reveals that the thin sheet-like layered structure of the 2D nanosheets is retained after

the liquid exfoliation process (Fig. 3e; Supplementary Fig. 13d, e). Interestingly, although the NUS-24 bulk powder is mainly amorphous based on the PXRD result, we can still observe clear crystal lattice in the exfoliated NUS-24 nanosheets by HR-TEM, indicating that the NUS-24 has ordered orientations in few-layered microstructures (Fig. 3f, g). In order to fully study the possible long-range orders, we optimized 2D layered crystalline structures of AA stacking and AB stacking of NUS-24 with the density functional tight-binding (DFTB+) method[49] incorporating dispersion interactions (Supplementary Note 2). Although the pore size distribution of AA stacking model (20.2

Å × 36.9 Å) and AB stacking model (9.3 Å × 14.0 Å) do not agree well with the experimental data of NUS-24 bulk (Supplementary Figs. 14, 15; Supplementary Tables 1–4), the total stacking energy of AA stacking (540.2 kcal mol$^{-1}$) is much lower than that of the AB stacking (1831.5 kcal mol$^{-1}$). In addition, after exfoliation, the pore size distribution of NUS-24 nanosheets only contain mesopores (around 28 Å), which is close to AA stacking model. Based on the above results, we believe that NUS-24 may adopt AA stacking, although the possibility of slipped stacking cannot be completely ruled out. With the aid of AA stacking model, it is shown that the two adjacent TPE rotors is around 3.7 Å apart between two layers (Supplementary Fig. 14e), and are attracted through π−π stacking that restricts their possible motions. Based on this model, it may be concluded that the lattice fringe spacing of 0.45 nm measured from the HR-TEM of NUS-24 nanosheets originates from the (660) crystal plane of the optimized structure (Fig. 3h). Similar phenomena have been observed in other 1D polymers[50] and 2D polymers[51,52].

The 2D layered structure of NUS-24 was further confirmed by atomic force microscopy (AFM) investigation. The NUS-24 nanosheets exfoliated in acetone were transferred on silicon wafers and studied by AFM for their lamellar features (Fig. 3i, j). As demonstrated by the AFM analyses on a total of 322 sites (Supplementary Fig. 16; Supplementary Note 3), more than 90% of the exfoliated NUS-24 have sheet-like morphology with a thickness of 2–5 nm (Fig. 3m) and a rather broad lateral size distribution (1.0–10.0 μm), indicating three to six layers of NUS-24 nanosheets based on the optimized AA stacking modeling structure (Supplementary Fig. 17). On the other hand, exfoliation in acetonitrile and ethanol can also be easily identified with the thickness of 3–5 nm (Fig. 3k, l). Notably, the exfoliation and stabilization of NUS-24 nanosheets can be facilitated by the dynamics of the TPE rotors, which helps to weaken the interlayer π−π stacking and prevent the re-stacking of exfoliated nanosheets. Even after 60 days, NUS-24 nanosheets suspended in acetone, acetonitrile, or ethanol solutions still exhibited a homogeneous state with a thickness of 3–4 nm (Supplementary Fig. 18). Dynamic light scattering (DLS) tests indicate that the average size of these nanosheets is 529 nm in acetone, 404 nm in acetonitrile, and 657 nm in ethanol based on the Stokes–Einstein equation[53] (Fig. 3n). In addition, the clear Tyndall effect of solutions containing exfoliated NUS-24 provides strong evidence for the colloidal ultrathin 2D nanosheets (Fig. 3o). Moreover, the nanosheets exhibit stronger fluorescence emission at 77 K compared to that at room temperature (Fig. 3p), suggesting the turn-on fluorescence emission caused by the freezing of TPE rotors at low temperatures. This phenomenon is reminiscent of aggregation-induced emission (AIE), in which turn-on fluorescence emission can be obtained by the restriction of molecular motions[54,55]. Our expectation was further proven by cryogenic differential scanning calorimetry (DSC). A distinct endothermic peak at −80 °C was observed in bulk NUS-24 during the heating scan (Supplementary Fig. 19), indicating a phase transition in which the frozen TPE molecular rotors become rotatable upon heating. Similar phase transitions have been observed in our previously reported MOF (NUS-1)[23] and POF (NUS-22)[22], suggesting the presence of highly dynamic TPE molecular rotors in the framework of NUS-24. The above characterization results strongly indicate the 2D layered structure of NUS-24, which can be exfoliated into nanosheets with liberated TPE molecular rotors that are suitable for further applications.

**Dynamic behavior of TPE rotors in NUS-24 nanosheets.** To have a better understanding of the dynamic behavior of the TPE rotors in NUS-24 nanosheets, the AIE characteristics of TPE-1,

TPE-2, NUS-24 bulk powder and NUS-24 nanosheets were studied in mixtures of acetone and water with different water fractions ($f_w$). As expected, TPE-1 and TPE-2 linkers exhibit typical AIE characteristics. The highest fluorescence intensities were obtained at the $f_w$ of 90%, and are 1386- and 2607-fold higher than that in acetone solutions for TPE-1 and TPE-2, respectively (Supplementary Fig. 20a–d). The AIE characteristics of NUS-24 bulk powder and nanosheets are not as obvious as that of TPE-1 and TPE-2, which is mainly because of the restriction of TPE rotors in the polymeric networks. However, we can still observe gradual fluorescent enhancement with the increase of $f_w$ (Fig. 4a; Supplementary Fig. 20e). Interestingly, we found that the fluorescence enhancement of NUS-24 nanosheets became faster at $f_w$ > 40% (Fig. 4b), which can be attributed to the restriction of TPE rotors by re-stacking of nanosheets in solutions with higher $f_w$. The fluorescence emission of NUS-24 bulk powder also gradually enhanced at higher $f_w$ (Supplementary Fig. 20f), while the sensitivity is lower than that of the nanosheets. At $f_w$ of 90%, the fluorescence emissions of NUS-24 nanosheets and bulk powder are 1.87- and 1.26-fold higher than that in pure acetone solutions, respectively. This result clearly indicates the importance of dynamic TPE rotors exposed on the surface of the nanosheets in enhancing the sensitivity of AIE characteristics. Similar phenomenon was also observed in flexible polymers containing large free volumes of TPE unit for AIE features[56].

Second, the dynamics of TPE rotors was further demonstrated by size-selective VOC sensing. The experiments were performed by soaking NUS-24 nanosheets in various VOCs of different molecular size followed by photoluminescence tests. As expected, the NUS-24 nanosheets exhibit turn-on fluorescence after being soaked in VOCs, with rough positive correlations between the emission intensity of NUS-24 nanosheets and the molecular size of VOCs. For example, the NUS-24 nanosheets exhibit a yellow emission in acetone (Fig. 4c), and the emission intensity increases with increasing molecular size of the VOC analytes, including dichloromethane, chloroform, tetrahydrofuran, n-hexane, cyclohexane, benzene, toluene, o-xylene, m-xylene, p-xylene, and 1,4-diisopropylbenzene (Supplementary Fig. 21; Supplementary Note 4). Using the emission intensity of NUS-24 nanosheets in acetone ($I_{acetone}$) as the reference, the relative intensities ($I_R = I/I_{acetone}$) of NUS-24 nanosheets in other analytes are shown in Fig. 4e. As expected, 1,4-diisopropylbenzene (4.3 × 9.4 Å), which is the largest analyte, has the highest value of $I_R$ (4.40) among all the tested VOCs. Compared to NUS-24 nanosheets, the emission intensity increase of NUS-24 bulk powder soaking in VOCs is relatively weaker (Supplementary Fig. 22). This suggests the higher sensitivity of NUS-24 nanosheets compared to that of NUS-24 bulk powder, and this should originate from the stronger interactions of liberated TPE rotors on the surface of NUS-24 nanosheets with analytes. It is worth noting that the different $I_R$ values between nanosheets⊃benzene (2.11) and nanosheets⊃toluene (2.61) suggest that NUS-24 nanosheets can be used to discriminate benzene from toluene. This is of great importance in VOC sensing due to the different toxicity of these two compounds[57]. In addition, NUS-24 nanosheets also demonstrate a good discrimination among xylene isomers (o-xylene/m-xylene/p-xylene), which is currently a big challenge in chemical sensing due to their similar structures and physicochemical properties[58]. For example, the $I_R$ values of nanosheets⊃o-xylene, nanosheets⊃m-xylene, and nanosheets⊃p-xylene are 3.24, 2.80, and 3.77, respectively. On the contrary, NUS-24 bulk powder exhibits poor discrimination as $I_R$ values of bulk⊃o-xylene (1.29), bulk⊃m-xylene (1.28), and bulk⊃p-xylene (1.34) indicate almost the same emission intensity (Fig. 4e). The control experiment indicates that the TPE-1 linker is unable to detect any VOCs (Supplementary Fig. 23), mainly because it is almost non-emissive

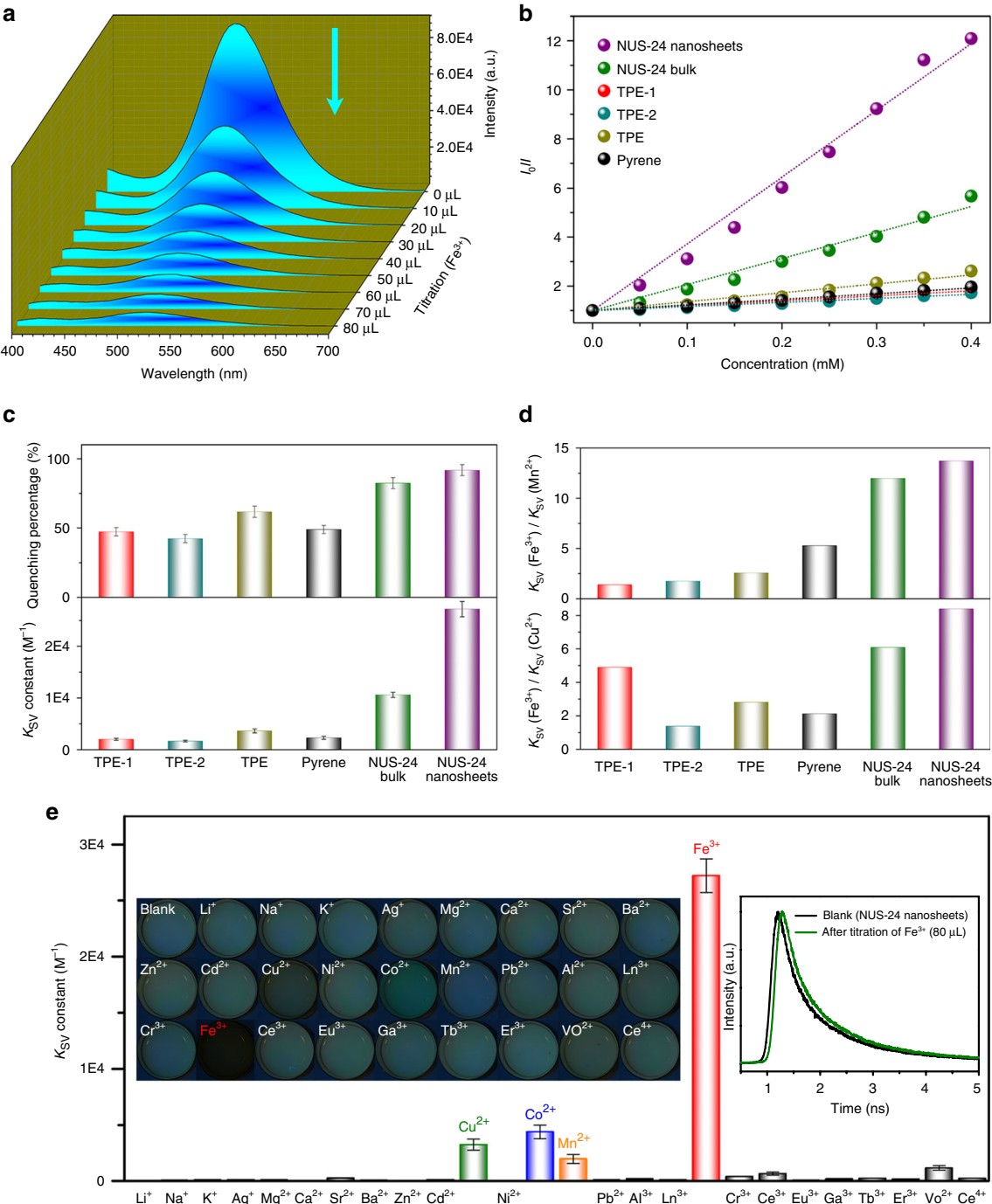

**Fig. 5** Chemical sensing of metal ions by NUS-24 nanosheets. **a** Fluorescence emission spectra of NUS-24 nanosheets ($c = 0.1$ mg mL$^{-1}$) upon titration with Fe$^{3+}$ solution ($1 \times 10^{-2}$ M) at room temperature ($\lambda_{ex} = 365$ nm). **b** Stern–Völmer plots of NUS-24 nanosheets, bulk powder, and compared small molecules being titrated with Fe$^{3+}$. **c** The quenching percentages and $K_{sv}$ constants of NUS-24 nanosheets, bulk powder, and compared small molecules by Fe$^{3+}$. **d** The ion selectivity of NUS-24 nanosheets, bulk powder, and compared small molecules. **e** The $K_{SV}$ constants of NUS-24 nanosheets being titrated with different metal ion solutions (left inset: fluorescent photograph of NUS-24 nanosheets upon titration with different metal ion solutions, $\lambda_{ex} = 365$ nm; Right inset: emission decay trace of NUS-24 nanosheets before and after the addition of Fe$^{3+}$ solution at room temperature). The error bars in **c** and **e** show average and range of measured values based on at least five repeats in every single measurement

in VOCs. We believe that the exposed TPE rotors on the large surface of NUS-24 nanosheets create a more discriminating environment than the free TPE linkers, resulting in much enhanced sensitivity and selectivity in chemical sensing.

Moreover, there is no obvious fluorescence peak shift in the turn-on process for analytes ranging from acetone to 1,4-diisopropylbenzene, suggesting no strong π−π stacking between NUS-24 nanosheets and VOC analytes. Therefore, the VOC sensing principle of NUS-24 nanosheets should mainly follow the AIE mechanism[54]. Essentially, the fluorescence emission of NUS-24 nanosheets is positively proportional to the restriction of TPE rotors caused by interacting with VOC analytes. Larger

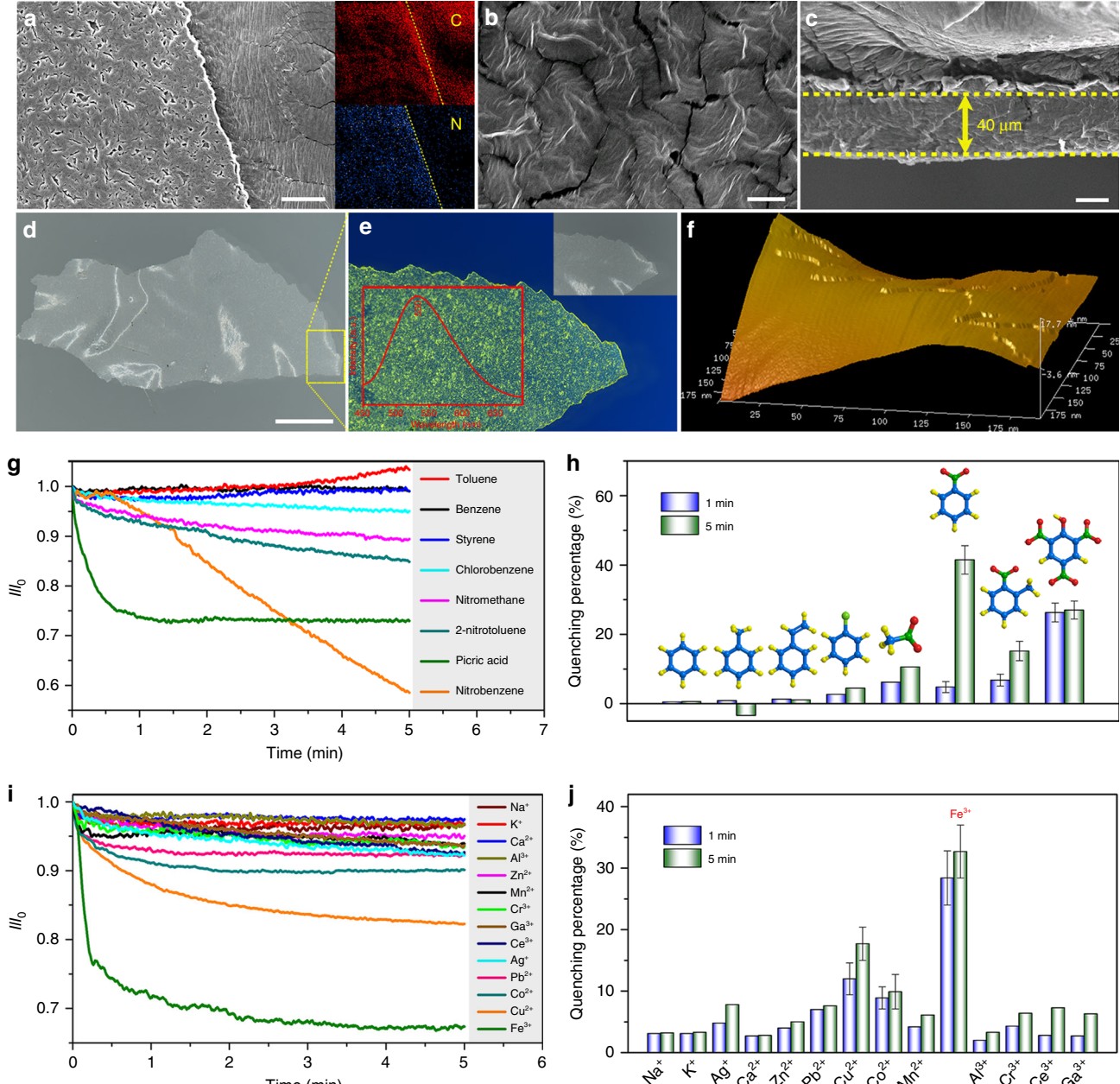

**Fig. 6** Chemical sensing by MMMs containing NUS-24 nanosheets. **a**, **b** FE-SEM and EDX elemental mapping (carbon and nitrogen) images of the MMMs (scale bar, 50 μm in **a** and 5 μm in **b**). **c** Cross-sectional FE-SEM image of the MMMs (scale bar, 20 μm). **d**, **e** Optical (**d**) and fluorescence (**e**) photographs of the MMMs (scale bar, 5 mm in **d**. $\lambda_{ex} = 365$ nm). **f** AFM image of the MMMs. **g** Time-dependent fluorescence intensity changes ($I/I_0$) of the MMMs upon 5 min exposure to VOCs vapors ($\lambda_{ex} = 365$ nm, $\lambda_{em} = 530$ nm). **h** The quenching percentages of the MMMs by VOC vapors after 1 and 5 min exposure. **i** Time-dependent fluorescence intensity changes ($I/I_0$) of the MMMs upon 5 min exposure to metal ion solutions ($\lambda_{ex} = 365$ nm, $\lambda_{em} = 530$ nm). **j** The quenching percentages of the MMMs by metal ions after 1 and 5 min exposure. The error bars in **h** and **j** show standard deviations based on three independent measurements

analytes tend to interact more strongly with TPE rotors, resulting in stronger fluorescence emissions[59]. Compared to the TPE rotors in 3D porous materials[22,23], the TPE rotors in NUS-24 nanosheets exhibit dynamic time-dependent behavior when interacting with VOC analytes. As can be seen in Fig. 4d, 15 min is required for the highest fluorescence emission to be reached after adding 20% of 1,4-diisopropylbenzene into acetone suspension containing NUS-24 nanosheets (Supplementary Fig. 24). This phenomenon is probably due to the weaker confinement effect of 2D nanosheets exerted onto the VOC

analytes compared to that of the 3D porous materials. Hence, a longer time is required to reach equilibrium where maximum restriction on the dangling TPE rotors, as well as the highest fluorescence emission, is achieved. Moreover, the static nature of the turn-on process is also proven by consistent fluorescence lifetimes of NUS-24 nanosheets before and after adding VOCs. For example, the lifetimes ($\tau_0$) of nanosheets⊃acetone, nanosheets⊃$n$-hexane, nanosheets⊃benzene, and nanosheets⊃1,4-diisopropylbenzene were calculated to be 1.04, 1.11, 1.11, and 1.12 ns, respectively, by the biexponential fitting of

the fluorescence emission decay data (Supplementary Fig. 25). Besides turn-on, the turn-off mode was also observed in NUS-24 nanosheets using nitrobenzene as the analyte, which is well known for its ability in quenching fluorescence[60]. We could clearly observe a huge blue shift (~60 nm) of the fluorescence peak from nanosheets (520 nm) to nanosheets⊃nitrobenzene (460 nm) (Fig. 4c), indicating that the turn-off process should be mainly caused by donor–acceptor electron transfer mechanism[61] instead of AIE mechanism observed in the turn-on process (Supplementary Fig. 26; Supplementary Note 5).

To further investigate the effect of analyte's molecular size on the restriction of dynamic TPE rotors, fluorescence titrations were carried out with gradual addition of small amount of benzene (4.3 × 5.0 Å), naphthalene (5.0 × 6.7 Å), or phenanthrene (5.0 × 9.2 Å) to the acetone suspensions containing NUS-24 nanosheets (Supplementary Fig. 27). As expected, fluorescence enhancement was also observed following the order of phenanthrene > naphthalene > benzene, agreeing well with the size effect elucidated previously. This study gives a rare example of ultrathin 2D nanosheets containing molecular rotors with almost perfect linear relationship between turn-on fluorescence and analyte concentration (Supplementary Note 6).

**Chemical sensing of metal ions by NUS-24 nanosheets.** In order to get more insights into the chemical sensing behavior of NUS-24 nanosheets, we also screened their capability in the chemical sensing of metal ions including transition metal ions ($Ag^+$, $Mn^{2+}$, $Co^{2+}$, $Ni^{2+}$, $Cu^{2+}$, $Zn^{2+}$, $Cd^{2+}$, $Pb^{2+}$, $Al^{3+}$, $Ga^{3+}$, $Fe^{3+}$, $Ln^{3+}$, $Cr^{3+}$), alkali metal ions ($Li^+$, $Na^+$, $K^+$), alkaline earth metal ions ($Mg^{2+}$, $Ca^{2+}$, $Ba^{2+}$, $Sr^{2+}$), and rare earth metal ions ($Ce^{3+}$, $Eu^{3+}$, $Tb^{3+}$, $Er^{3+}$). Among these metal ions, $Cu^{2+}$, $Co^{2+}$, $Mn^{2+}$, and $Fe^{3+}$ exhibit strong quenching effects (Supplementary Fig. 28). In particular, the emission intensity of NUS-24 nanosheets keeps decreasing with gradual addition of $Fe^{3+}$ (Fig. 5a), affording a linear relationship ($R^2 = 0.9959$), which agrees with the Stern–Völmer (S–V) equation[62] and suggesting a 1:1 stoichiometry of chemical sensing (Fig. 5b). $Fe^{3+}$ demonstrates the highest quenching percentage of 91.7%, while the quenching percentages of $Cu^{2+}$, $Co^{2+}$, and $Mn^{2+}$ are 54.1%, 51.1%, and 41.4% (Supplementary Fig. 29), respectively. The quenching constant $K_{SV}$ represents the quenching sensitivity[62,63], and the $K_{SV}$ values were calculated to be 27,214, 4387, 3242, and 1984 $M^{-1}$ for $Fe^{3+}$, $Co^{2+}$, $Cu^{2+}$, and $Mn^{2+}$, respectively. The quenching percentage of other metal ions are 2–30% and the $K_{SV}$ are 30–1200 $M^{-1}$ (Fig. 5e; Supplementary Table 6), which are all much lower than that of $Fe^{3+}$. This is probably because $Fe^{3+}$ contains unoccupied $d$ orbitals and has a smaller diameter compared to other metal ions[64]. Therefore, when the nanosheets interact with $Fe^{3+}$, the electrons can easily transfer from nanosheets to $Fe^{3+}$, leading to quick fluorescent quenching of the nanosheets.

In order to demonstrate the advantages of nanosheets for $Fe^{3+}$ sensing, we chose several fluorescent molecules including TPE-1, TPE-2, TPE, and pyrene as well as NUS-24 bulk powder for comparison (Supplementary Fig. 30). Clearly, the NUS-24 nanosheets can be quenched faster than the small fluorescent molecules and NUS-24 bulk powder (Fig. 5b; Supplementary Figs. 31–35). The $K_{SV}$ values were calculated to be 27,214, 10,617, 3632, 2296, 2011, and 1659 $M^{-1}$ for NUS-24 nanosheets, NUS-24 bulk powder, TPE, pyrene, TPE-1, and TPE-2, respectively (Fig. 5c). In addition, three other metal ions including $Cu^{2+}$, $Mn^{2+}$, and $Co^{2+}$ were also tested. Again, NUS-24 nanosheets display higher sensitivity compared to its bulk powder as well as the small fluorescent molecules in sensing those metal ions (Supplementary Fig. 36; Supplementary Table 7). The superior

sensitivity of NUS-24 nanosheets can be attributed to their fully exposed external surface area allowing sufficient contact and interaction with analytes. Finally, ion selectivity was determined using $K_{sv}(Fe^{3+})/K_{sv}(Mn^{2+})$ and $K_{sv}(Fe^{3+})/K_{sv}(Cu^{2+})$. As shown in Fig. 5d, NUS-24 nanosheets show a much higher selectivity toward $Fe^{3+}$ than the four small fluorescent molecules as well as the bulk powder (Supplementary Table 8), indicating the advantage of nanosheets for metal ion sensing.

Inspired by the high sensitivity and selectivity of NUS-24 nanosheet toward $Fe^{3+}$, which plays a significant role in many biochemical processes such as oxygen metabolism and synthesis of DNA and RNA[65], we conducted a detailed study on the trace amount sensing of $Fe^{3+}$ using NUS-24 nanosheets. The apparent quenching constant $K_q$ ($K_q = K_{SV}/\tau_0$)[66] for NUS-24 nanosheets quenched by $Fe^{3+}$ was evaluated to be $2.62 \times 10^{13}$ $M^{-1}$ $s^{-1}$, which is three orders of magnitude higher than that of conventional bimolecular quenching systems (~$10^{10}$ $M^{-1}$ $s^{-1}$)[67]. In addition, NUS-24 nanosheets can detect $Fe^{3+}$ at very low concentrations by fluorescence quenching (Supplementary Figs. 37, 38). The detection limit[68] ($3\sigma/K_{SV}$, $\sigma$ is the standard deviation of this detection method) of $Fe^{3+}$ was calculated to be $9 \times 10^{-4}$ M. In order to further understand the quenching details, this process was examined by time-resolved fluorescence measurements. The lifetimes of NUS-24 nanosheets remained almost unchanged before and after adding $Fe^{3+}$ (1.04 versus 1.09 ns, the right inset of Fig. 5e), suggesting a static nature of the fluorescence quenching. Furthermore, the fluorescence quenching of NUS-24 nanosheets by $Fe^{3+}$ can be easily seen from the fluorescence microscopy images (the left inset of Fig. 5e), implying the potential applications of NUS-24 nanosheets in naked eye detection of $Fe^{3+}$.

**Mixed matrix membranes containing NUS-24 nanosheets.** We incorporated NUS-24 nanosheets into mixed matrix membranes (MMMs) in order to further understand the effect of TPE rotors in practical sensing applications, especially the turn-on fluorescence for VOC sensing. We hypothesize that the TPE rotors in the free-standing NUS-24 nanosheets can be restricted by the polymeric matrix in MMMs, leading to attenuation or even loss of turn-on fluorescence in the presence of VOCs. We chose poly(ethylene imine) (PEI) as the polymeric matrix for the preparation of MMMs because of the lack of electron-deficient groups that minimizes interference with the fluorescence emission of NUS-24 nanosheets. MMMs were fabricated by casting stock solutions containing fully dissolved polymers and suspended NUS-24 nanosheets[69]. FE-SEM, energy-dispersive X-ray spectroscopy (EDX) elemental mapping, and XPS spectra indicate the pure organic composition of MMMs containing only carbon and nitrogen elements (Fig. 6a, b; Supplementary Figs. 39–41). The thickness of the membrane was estimated to be around 40 µm from the cross-sectional SEM image (Fig. 6c). A flat and smooth surface of the membrane was confirmed by AFM (Fig. 6f). The uniform yellow fluorescence emission of the membrane suggests a homogeneous dispersion of NUS-24 nanosheets within the PEI matrix (Fig. 6d, e).

In order to test our previous hypothesis, we exposed MMMs in various VOC vapors (benzene, toluene, styrene, and chlorobenzene) and checked their fluorescence emission ($\lambda_{ex} = 365$ nm, $\lambda_{em} = 530$ nm). As has been expected, there was barely any turn-on fluorescence being observed after 5 min exposure to VOC vapors (fluorescence enhancement <5%, Fig. 6g, h), confirming the restriction of TPE rotors by PEI matrix, which leads to reduced sensitivity toward VOC sensing based on turn-on fluorescence. On the other hand, it is possible that the polymeric matrix works as a barrier to slow down the diffusion of VOC molecules leading to poor interactions with NUS-24 nanosheets and accordingly weaken turn-on fluorescence. To rule out this possibility, we

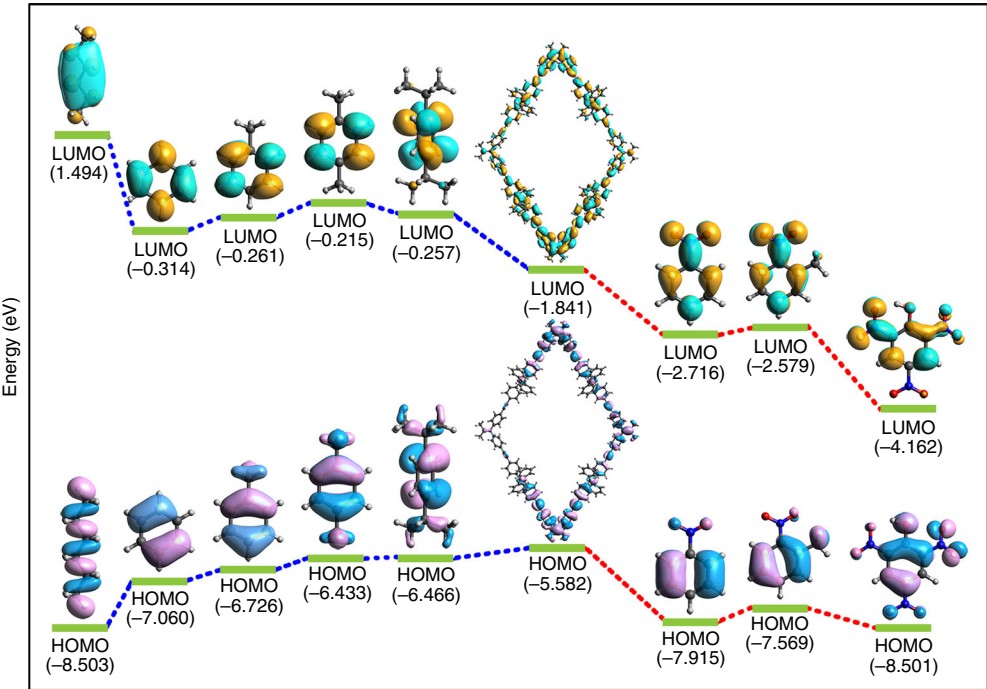

**Fig. 7** DFT-calculated HOMO–LUMO energy profiles. HOMO–LUMO energy profiles of *n*-hexane, benzene, toluene, *p*-xylene, 1,4-diisopropylbenzene, NUS-24 fragment, nitrobenzene, 2-nitrotoluene, and 2,4,6-trinitrophenol (from left to right)

exposed MMMs in the vapors of fluorescence quenching compounds including nitromethane, nitrobenzene, 2-nitrotoluene, and 2,4,6-trinitrophenol (picric acid). Dramatic fluorescence quenching of MMMs was observed in 1 min, indicating that the barrier effect imposed by PEI matrix, if there is any, should not be the major reason for the attenuated turn-on fluorescence in MMM-based VOC sensing.

Interestingly, we noticed that 2,4,6-trinitrophenol exhibited the fastest quenching rate within the initial 1 min among all the fluorescence quenching compounds (Fig. 6g, h). This can be attributed to its three electron-withdrawing nitro groups that strongly quench the fluorescence via donor–acceptor electron transfer mechanism. However, the quenching process caused by 2,4,6-trinitrophenol saturated after 1 min, which may be due to the hydrogen bonding interaction between the -OH groups of 2,4,6-trinitrophenol and the -NH- groups of PEI, preventing further penetration of 2,4,6-trinitrophenol into the membranes. On the contrary, nitrobenzene, with a smaller molecular size and lack of hydrogen bonding interaction, could continuously quench the fluorescence emission of MMMs over the whole test period (5 min). After 5 min, the quenching percentage obtained using nitrobenzene is 41.5%, which is much higher than that obtained using 2,4,6-trinitrophenol (27.0%), 2-nitrotoluene (15.2%), and nitromethane (10.6%). These results suggest that MMMs containing NUS-24 nanosheets can be used for the detection of explosives, which are mainly nitro-containing compounds[60]. We further used the MMMs for the chemical sensing of metal ions in aqueous solutions. As shown in Fig. 6i, $Fe^{3+}$ also exhibits the fastest quenching speed among all the metal ions being tested. After 1 min of soaking MMMs in metal ion solutions, the quenching percentages are 28.4, 12.0, and 8.9% for $Fe^{3+}$, $Cu^{2+}$, and $Co^{2+}$, respectively, while the other metal ions are all <7% (Fig. 6j).

**DFT calculations**. DFT calculations were performed on NUS-24 and selected analytes to better understand the different fluorescence emission behavior such as 1,4-diisopropylbenzene

(turn-on) and nitrobenzene (turn-off). DFT calculations (Fig. 7) show that the LUMO of an NUS-24 fragment (−1.841 eV) lies higher in energy than the LUMO of nitrobenzene (−2.716 eV), 2-nitrotoluene (−2.579 eV), and 2,4,6-trinitrophenol (−4.162 eV). Therefore, efficient electron transfer may occur from NUS-24 to these nitro compounds, leading to fluorescence quench[22,37,61]. In contrast, the LUMOs of electron-rich VOC analytes (e.g., *n*-hexane, benzene, toluene, *p*-xylene, and 1,4-diisopropylbenzene) are higher-lying than NUS-24 fragment. As a result, electrons may transfer from electron-rich analytes to NUS-24, which should enhance the NUS-24 fluorescence[61,62]. However, compared to the electron transfer mechanism, the AIE mechanism (i.e., the restriction of TPE rotors by interacting with analytes) should dominate the turn-on fluorescence of NUS-24 nanosheets in the presence of VOCs and this has been proven by the results of MMMs sensing.

## Discussion

In summary, we have introduced TPE molecular rotors into 2D PONs (NUS-24) for fluorescence-based chemical sensing. Compared to NUS-24 bulk powder, the exfoliated NUS-24 nanosheets exhibit stronger turn-on fluorescence upon contact with electron-rich VOCs, and the fluorescence enhancement is positively correlated with the molecular size of VOC molecules. This can be attributed to the restriction of liberated TPE rotors on the external surface of NUS-24 nanosheets when interacting with VOC molecules, which is similar to the AIE mechanism. The proposed mechanism was further proven by incorporating NUS-24 nanosheets into PEI affording MMMs, by which the TPE rotors can be almost completely restricted by the polymeric matrix resulting in greatly attenuated turn-on fluorescence upon exposure to VOC vapors. Nevertheless, the NUS-24 nanosheets and MMMs exhibit practical sensing capability toward nitro-containing compounds and $Fe^{3+}$ ion through fluorescence quenching caused by donor–acceptor electron transfer mechanism, which is proven by DFT calculations. Our study has thus opened a new door toward the rational design of smart 2D

materials containing molecular rotors, and demonstrated their promising applications in explosive detection and environmental monitoring.

## Methods

**Synthesis of NUS-24 bulk powder.** Syntheses of TPE-1 and TPE-2 linker were performed according to previously reported procedures (see Supplementary Methods for more details). NUS-24 bulk powder was synthesized using Sonogashira–Hagihara coupling reactions. Briefly, a mixture of TPE-1 linker (196 mg, 0.4 mmol), TPE-2 linker (86 mg, 0.2 mmol), tetrakis(triphenylphosphine) palladium (11.6 mg, 0.01 mmol), and copper(I) iodide (38 mg, 0.2 mmol) in $N,N$-dimethylformamide/triethylamine (8 mL/8 mL) was degassed and purged with nitrogen. The mixture was stirred at 90 °C for 72 h and then cooled to room temperature before being poured into water. The precipitate was collected by filtration, repeatedly rinsed with hydrochloric acid (2 M), water, tetrahydrofuran, ethanol, dichloromethane, acetone, and then rigorously washed by Soxhlet extraction for 24 h with chloroform, tetrahydrofuran, and acetone sequentially, and finally dried in vacuum to give NUS-24 bulk powder (187.5 mg, 84% yield) as deep yellow powder.

**Preparation of NUS-24 nanosheets.** NUS-24 nanosheets were prepared from NUS-24 bulk powder by ultrasonic exfoliation in organic solvents. Briefly, 10 mg of the NUS-24 bulk powder sample was suspended in 10 mL of solvent (acetone, acetonitrile, or ethanol), and was sonicated with a frequency of 40 kHz for 6 h to give a homogeneous dispersion. The resulting dispersion was centrifuged at 3500 rpm for 5 min. The supernatant was collected and followed by another centrifugation at 8000 rpm for 10 min to further remove non-exfoliated powder.

**Fabrication of MMMs.** In a typical process, acetone suspension (2.5 mL) containing exfoliated NUS-24 nanosheets (0.1 mg mL$^{-1}$) was sonicated for 30 min using an ultrasonic homogenizer (Biobase, JY92-IIDN), followed by stirring for another 120 min. This cycle was repeated three times, and then PEI (200 mg) dissolved in ethanol (2.5 mL) was added followed by another sonication-stirring cycle to give the membrane casting solution, which was casted onto a flat glass substrate followed by slow vaporization of the solvent to give the final MMMs.

**Chemical sensing of VOCs and metal ions.** The acetone suspension (40 μL) containing exfoliated NUS-24 nanosheets (0.1 mg mL$^{-1}$) was added into individual VOC solution (2 mL), which was thoroughly stirred before each photoluminescence measurement. Fluorescence spectra were recorded on a PTI/QM spectrophotometer. The excitation wavelength for liquid VOC sensing was 365 nm. Titration experiments of metal ions were carried out by adding aliquots (10 μL) of metal salt solutions ($1.0 \times 10^{-2}$ M) into the acetone suspension (2 mL) containing NUS-24 nanosheets (0.1 mg mL$^{-1}$) at intervals of 5 min. Fluorescence spectra were recorded after the addition of metal salt solutions. The excitation wavelength was 365 nm. The fluorescence quenching was analyzed using the Stern–Völmer equations derived for 1:1 complexes to determine the binding mode:

$$I_0/I = 1 + K_{sv}[Q]. \tag{1}$$

The procedure for the chemical sensing of VOC vapors and metal ions using MMMs is as follows: The MMM ($1 \times 1$ cm) was fixed into the inner surface of a quartz cuvette, which was placed into the saturated VOC vapors or metal salt solutions (2 mL, $1 \times 10^{-3}$ M in aqueous solution) for 5 min followed by photoluminescence test ($\lambda_{ex} = 365$ nm, $\lambda_{em} = 530$ nm). The quenching percentage was estimated using the formula $(I_0−I)/I_0 \times 100\%$, where $I_0$ is the original maximum peak intensity and $I$ is the maximum peak intensity after exposure to VOC vapors or metal ion solutions.

**Energy barrier calculations of TPE rotor.** To examine the role of free phenyl groups of TPE-1 linker, 37 cluster models were constructed containing one TPE-1 linker and two TPE-2 linkers. The dangling bonds in the cluster models were saturated by H atoms. The rotor angle between the C–C bonds between two free phenyl groups varied from 0° to 360° with an interval of 10°. The constraint optimization was carried out using the B3LYP functional with 6-31G(d) basis set, followed by the single point energy calculation at M062X/6-31+G(d) level in a solvent (acetone)[70]. The solvent was represented by a polarizable continuum model[71] and the thermal correction to Gibbs free energy was estimated at 298 K. All the DFT calculations were performed by Gaussian 09.

**HOMO–LUMO energy calculations.** To quantitatively evaluate the interactions between NUS-24 and VOC molecules, the electronic properties of a NUS-24 fragment and eight VOC molecules, i.e., $n$-hexane, 1,4-diisopropylbenzene, $p$-xylene, toluene, benzene, 2-nitrotoluene, nitrobenzene, and 2,4,6-trinitrophenol, were calculated using DFT. Initially, the simulated structure of NUS-24 was optimized by Forcite using Materials Studio to remove geometric distortions.

Then, an NUS-24 fragment with a complete ring containing all typical building blocks was used in DFT calculations. The cleaved bonds of the ring were terminated by hydrogen atoms. The VOC molecules were optimized using the B3PW912 hybrid functional with 6-31G(d) basis set. The HOMO and LUMO energy levels of the NUS-24 fragment and VOC molecules were calculated using the B3PW91 hybrid function with 6-311 G** basis set. The B3PW91 function was developed by Becke[72] and demonstrated to be computationally accurate and fast for band gap calculations[73]. All the DFT calculations were carried out using Gaussian 09.

**Characterization.** Nuclear magnetic resonance spectroscopy (NMR) data were collected on a Bruker Avance 400 MHz NMR spectrometer (DRX400). FTIR spectra were obtained with a Bio-Rad FTS-3500 ARX FTIR spectrometer. UV–Vis spectra were collected in the solid state on a Shimadzu UV-3600 spectrometer using the BaSO$_4$ reflectance standard at room temperature. XPS experiments were performed with a Kratos AXIS Ultra DLD surface analysis instrument using a monochromatic Al Kα radiation (1486.71 eV) at 15 kV as the excitation source. PXRD patterns were obtained on a Rigaku MiniFlex 600 X-ray powder diffractometer equipped with a Cu sealed tube ($\lambda = 1.54178$ Å) at a scan rate of 2° per min. TGA were performed using a Shimadzu DTG-60AH in the temperature range of 100–750 °C under flowing air (50 mL min$^{-1}$) and a heating rate of 10 °C per min. DSC analyses were carried out with a Mettler Toledo DSC822e DSC under N$_2$ atmosphere with a cooling/heating rate of 20 °C per min. FE-SEM was conducted on a JEOL JSM-7610F scanning electron microscope. Samples were treated via Pt sputtering for 90 s before observation. TEM was conducted on a JEOL JEM-3010 transmission electron microscope. AFM was conducted by testing samples deposited on silica wafers using tapping mode with a Bruker Dimension Icon atomic force microscopes. N$_2$ sorption isotherms were measured using a Micromeritics ASAP 2020 surface area and pore size analyzer. Before the measurements, the samples were degassed under high vacuum (<0.01 Pa) at 120 °C for 10 h. UHP-grade helium and nitrogen were used for all the measurements. Fluorescence spectra were collected at room temperature on a Photon Technology International/QuantaMaster (PTI/QM, USA) spectrometer. Fluorescent microscopy images were acquired at an excitation wavelength of 365 nm using a Nikon Ti-U fluorescence microscope equipped with a 430 nm LP filter.

**Data availability.** The authors declare that all the data supporting the findings of this study are available within the article (and Supplementary Information Files), or available from the corresponding author on reasonable request.

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

## Acknowledgements

This work was supported by National University of Singapore (CENGas R-261-508-001-646), Ministry of Education—Singapore (MOE AcRF Tier 1 R-279-000-472-112), and Agency for Science, Technology and Research (PSF R-279-000-475-305).

## Author contributions

D.Z. formulated and supervised the project. J.D. synthesized NUS-24 bulk powder and prepared NUS-24 nanosheets, fabricated MMMs, performed spectroscopic characterization, morphology characterization, and chemical sensing. Y.Q. analyzed HR-TEM. K. Z., X.L., D.Y., and J.J. constructed the molecular models and conducted the DFT calculations. H.Z. and Q.-H.X. conducted the lifetime tests. J.D. and D.Z. wrote the paper, and all authors contributed to revising the paper.

## Additional information

**Competing interests:** D.Z. and J.D. declare the following competing financial interest(s): A SG non-provisional patent (No. 10201702327X) has been filed on 22 March 2017 based on the presented result. The remaining authors declare no competing financial interests.

