## [Peer Review File · Nature Communications]

Reviewers' comments:

Reviewer #1 (Remarks to the Author):

The manuscript by Dong et al reports the synthesis of a porous organic polymer named NUS 24 having tetraphenylethylene (TPE) as molecular rotor. Further, POP has been exfoliated to nanosheet by Liquid Phase Exfoliation and these nanosheets are shown to exhibit better performance in VOC and metal ion sensing in comparison to the bulk POP. The authors have carried out an interesting study. The manuscript can be accepted after addressing the following points:

1. The basic characterizations of nanosheet formation are missing from the manuscript. For example, no PXRD and N₂ sorption data of the nanosheet has been provided. However, these characterizations are very important in determining bulk to nanosheet formation. Authors are suggested to present the data of both the as-synthesized bulk material and the nanosheet of each characterization in one figure for comparison purpose.
2. Why the staking interaction amongst the layers is so poor, is not understood. Why the outer layer phenyl rings of the layers do not interact at all ?
3. The TEM image in Fig.S11 does not convey the nanosheet morphology properly. Further, TEM images of the bulk material are also required.
4. Authors must provide more info reg. the confirmation that the nanosheets are always five-layers, in particular when the material is amorphous in nature?
5. If the recognition of VOC are exclusively based on size, how do the authors justify the fluorescent quenching by nitrobenzene and aniline?
6. Why the nanosheet is particularly selective to Fe³⁺ among all the metal ions tested is not accounted in the manuscript. Further how the Fe³⁺ ion causes the PL quenching requires more clarifications.

Reviewer #2 (Remarks to the Author):

Recommendation: Resubmit after revisions.

Additional Comments:

Dong et al. report tetraphenylethylene (TPE) based n-conjugated 2D porous organic nanosheets named NUS-24 nanosheets for sensing VOCs and Fe³⁺. The materials have been well characterized and the sensing mechanisms have been deeply investigated. The research included in the manuscript could be of interest. However, this manuscript is too primary to publish. The following needed to be addressed before being matured for publication:

- 1) What's the advantage of NUS-24 nanosheets as compare with some small organic molecules based fluorescent sensor in sensing Fe³⁺. Indeed, the selectivity and sensitivity of NUS-24 is not that good.
- 2) The authors state that NUS-24 nanosheets can be used in environmental monitoring and biosensing applications of Fe³⁺, I think practical applications data such as imaging Fe³⁺ in living cells etc. should be provided.
- 3) AIE characteristics (in mixture of acetone and water) of TPE-1, TPE-2, NUS-24 bulk and NUS-24 nanosheets should be investigated and discussed.

Reviewer #3 (Remarks to the Author):

Zhou and coworkers present a detailed study of an interesting rhombic COF formed from planar 4-connected and 2-connected tetraphenylethylene (TPE) units. The COF is formed as an amorphous layered material due to the irreversible coupling reaction used, however, the authors are able to exfoliate the bulk powder to few-layer thick nanosheets. Screening studies show that the resultant nanosheets exhibit highly sensitive and selective sensing for VOCs and Fe³⁺.

I have some concerns about the modelling used. Page 9, line 168 states that five-layered nano sheets were formed based on the optimised structure and refers to Fig 1d. Fig 1d, however, is simply a schematic. Searching through the supplementary information for the structure, leads me to Fig S1 and Table S2. Firstly, these two pieces of data are not independent and should be presented together, also the cif file should be included in the SI. More importantly, I am not sure how the simulated parameters were arrived at. In particular, it seems to be that the c lattice parameter might have been chosen to agree with the AFM data. A true structural optimisation would stack layers together as tightly as possible - which is clearly not done here. I understand the difficulty of simulating solvent between layers, and do not ask the authors to do this, however, they do need to be upfront about the constraints applied to their simulation.

Following from this point, I would like to see modelling of the bulk NUS-24. Gross parameters such as density and indeed the pore size distribution could be matched and would yield information about the structural change (primarily freeing up the TPE rotors due to the increase in the c lattice constant) occurring upon exfoliation. When modelling the bulk, as layered MOFs/COFs exhibit slipped stacking due to interlayer repulsion and dispersion, I would expect the pore size to decrease somewhat, agreeing better with the 14 Å given by the bulk PSD.

On line 158, the authors state that their modelled structure is "very identical" to their previously published NUS-1. (Side note: it's either identical or it isn't). Are the authors sure about this? NUS-1 has a very interesting Kagome structure, which I see no evidence for in the present manuscript. I would think forming the NUS-1 structure would be quite difficult with the two given organic units. This needs to be cleared up.

Given the importance of the freed rotation of the 2-connected TPE units to the proposed mechanism of the framework, this point should also be supported by modelling. It would be quite straightforward for the authors to take a cutout of their 2-connected TPE linker and adjoining 4-connected units and rotate the two free phenyl groups, creating a 1D Potential Energy Surface for the rotation. Repeating this with implicit solvent and presumably seeing different barriers to rotation would support the role of the rotors in the sensing.

Finally, Fig S13 states that the sizes of the VOC molecules were estimated after optimisation in Materials Studio (without mention of method), however, in the main text, DFT calculations were undertaken on all these molecules. The DFT calculations should be used in preference and the details of the estimation provided (e.g. using the total density? vdW radius? In addition, if "square" dimensions are provided, the axes should be marked in the figure (e.g. from where to where corresponds to x Å measurement?).

July 15, 2017

Dr. Ariane Vartanian

Dear Dr. Vartanian,

Thank you very much for your feedback toward our manuscript entitled “**Ultrathin Two-Dimensional Porous Organic Nanosheets with Molecular Rotors for Chemical Sensing**” (Manuscript ID: NCOMMS-17-06757A-Z). We greatly appreciate the reviewers’ comments. Here are our point-to-point responses and changes made to the manuscript.

Reviewer #1 (Remarks to the Author):

The manuscript by Dong et al reports the synthesis of a porous organic polymer named NUS 24 having tetraphenylethylene (TPE) as molecular rotor. Further, POP has been exfoliated to nanosheet by Liquid Phase Exfoliation and these nanosheets are shown to exhibit better performance in VOC and metal ion sensing in comparison to the bulk POP. The authors have carried out an interesting study. The manuscript can be accepted after addressing the following points:

Response: We thank the reviewer for the positive comment of our work.

1. The basic characterizations of nanosheet formation are missing from the manuscript. For example, no PXRD and N₂ sorption data of the nanosheet has been provided. However, these characterizations are very important in determining bulk to nanosheet formation. Authors are suggested to present the data of both the as-synthesized bulk material and the nanosheet of each characterization in one figure for comparison purpose.

Response: We thank the reviewer for this comment. We strongly agree with this valuable suggestion. We have obtained the PXRD and N₂ sorption data of NUS-24 bulk powder, and put them with those of NUS-24 nanosheets in one figure (Figure R1). Compared to the NUS-24 bulk powder, the BET surface area of NUS-24 nanosheets decreases after exfoliation in different organic solvents such as acetone (100 m² g⁻¹), ethanol (162 m² g⁻¹), and acetonitrile (92 m² g⁻¹). Furthermore, the total pore volume of nanosheets (0.13, 0.20, and 0.13 cm³ g⁻¹ by exfoliation in acetone, ethanol, and acetonitrile, respectively) is also lower than that of the bulk powder (0.38 cm³ g⁻¹). This is mainly due to the disruption of π-π stacking in the layered structure of NUS-24 during the exfoliation process. Pore size distribution data indicate that the microporous structure of NUS-24 bulk powder (around 14 Å) almost disappears in the exfoliated 2D nanosheets. The PXRD patterns suggest that both bulk powder and nanosheets of NUS-24 have amorphous structures. All the above comments and data have been included in the revised manuscript (Supplementary Figures 10 and 11).

Figure R1. Comparison between NUS-24 bulk powder and NUS-24 nanosheets exfoliated in different organic solvents: (a) 77 K N₂ sorption isotherms (filled, adsorption; open, desorption); (b) Pore size distribution; (c) PXRD patterns.

2. Why the staking interaction amongst the layers is so poor, is not understood. Why the outer layer phenyl rings of the layers do not interact at all?

Response: We thank the reviewer for this comment. Here are our answers to the above two points:

(1) Compared to the synthesis of 2D crystalline covalent organic frameworks (COFs), the Sonogashira-Hagihara coupling reaction used for the synthesis of NUS-24 bulk polymer is irreversible, thereby leading to the generation of poor π - π stacking between the layers (please see the PXRD results). Furthermore, in this study, two twisted TPE linkers were employed to synthesize NUS-24 bulk polymer. The dynamic TPE rotors can also prevent the effective interlayer packing. Therefore, the highly distorted aromatic framework of NUS-24 with TPE rotors results in diminished interlayer interactions that favor the exfoliation and dispersion of bulk NUS-24 powder into 2D few-layered nanosheets in organic media. In this regard, our case is similar to the self-exfoliated ionic covalent organic nanosheets (*J. Am. Chem. Soc.* 2016, 138, 2823-2828) and exfoliable two-dimensional twisted conjugated microporous polymers (*Angew. Chem. Int. Ed.* 2017, 56, 6946-6951).

(2) Compared to the inner layer phenyl rings, the outer layer phenyl rings exposed on the surface of the ultrathin NUS-24 nanosheets should experience a much smaller π - π stacking interaction. To some extent, the outer layer phenyl rings are liberated from the confined configuration in the bulk powder, which is beneficial to fluorescence-based chemical sensing performance. For example, we have demonstrated that NUS-24 nanosheets are more sensitive and selective than their bulk powder in the chemical sensing of VOCs and metal ions. All the above comments have been added into the revised manuscript.

3. The TEM image in Fig.S11 does not convey the nanosheet morphology properly. Further, TEM images of the bulk material are also required.

Response: We thank the reviewer for this suggestion. The TEM analysis was conducted again for NUS-24 bulk powder and NUS-24 nanosheets (Figure R2). Compared to NUS-24 bulk powder, the NUS-24 nanosheets show thin, sheet-like layered structures, agreeing well with the AFM results. The discussion and data have been included in the revised manuscript (Main text Figure 3e and Supplementary Figure 13).

Figure R2. TEM images of NUS-24 bulk powder (a-c) and NUS-24 nanosheets (d-f).

4. Authors must provide more info reg. the confirmation that the nanosheets are always five-layers, in particular when the material is amorphous in nature?

Response: We thank the reviewer for this comment. According to this suggestion, we have measured 40 random places in the AFM images of five NUS-24 nanosheet samples (Figure

R3). The thicknesses of exfoliated NUS-24 nanosheets are summarized from the 322 AFM measurements (Figure R4). Based on the statistical analysis, 90% of the exfoliated NUS-24 nanosheets have sheet-like morphology with a thickness of 2 ~ 5 nm.

Figure R3. The 322 AFM thickness measurements of exfoliated NUS-24 nanosheets. Scale bars: 2 μm .

Figure R4. (a) Statistical analysis of the AFM thickness measurements. The theoretical height of NUS-24 nanosheets based on the AA stacking model: (b) three-layered (2.5 nm); (c) four-layered (3.3 nm); (d) five-layered (4.1 nm); (e) six-layered (4.9 nm).

Based on the suggestion from Reviewer 3, we have optimized the molecular modelling of NUS-24 by density-functional tight-binding (DFTB+) method. From DFTB+ method, we have calculated the theoretical height of three-layered (2.5 nm), four-layered (3.3 nm), five-layered (4.1 nm), and six-layered (4.9 nm) NUS-24 nanosheets. Therefore, it can be concluded that the exfoliated NUS-24 nanosheets in this study contain mainly three to six layers, with a majority of four layers.

Figure R5. The Tyndall effect and AFM images of exfoliated NUS-24 nanosheets in acetone (a-c, scale bar: 2 μm in Fig b), acetonitrile (d-f, scale bar: 0.5 μm in Fig e), and ethanol (g-i, scale bar: 1 μm in Fig h) after 60 days.

More interestingly, the as-exfoliated NUS-24 nanosheets remain stable in suspensions even after 60 days (Figure R5). This high stability may come from the dynamics of the TPE rotors, which help to weaken the interlayer π - π stacking and prevent the restacking of exfoliated nanosheets. The above discussion and data have been included in the revised manuscript (Main text Figure 3m and Supplementary Figures 16-18).

5. If the recognition of VOC are exclusively based on size, how do the authors justify the fluorescent quenching by nitrobenzene and aniline?

Response: We thank the reviewer for this comment. The recognition of VOCs in this study is based on the turn-on fluorescence caused by the restriction of TPE rotors, a mechanism that is identical to the aggregation-induced emission (AIE). However, if nitrobenzene (an aromatic compound with a strong electron-withdrawing nitro group) is used as the analyte, fluorescence quenching will be observed due to the donor-acceptor electron-transfer mechanism, which can overrule the previous AIE mechanism. This statement is further supported by the DFT calculation showing that the LUMO of the NUS-24 fragment (-1.841 eV) is higher in energy than the LUMO of nitrobenzene (-2.716 eV). Therefore, efficient electron transfer may occur from NUS-24 to nitrobenzene (please see Figure 7 in the main text), leading to fluorescence quenching. Our calculation results are consistent with the reported fluorescent MOFs whose emissions can also be quenched by nitrobenzene (*J. Am. Chem. Soc.* 2011, 133, 4153–4155). Since aniline always contains oxidation impurities such

as nitrobenzene, we have removed the aniline data in the revised manuscript to avoid misunderstanding.

Figure R6. (a) Fluorescence emission spectra of NUS-24 nanosheets ($c = 20 \mu\text{g/mL}$) upon titration with nitrobenzene solution ($1 \times 10^{-3} \text{ M}$) at room temperature ($\lambda_{\text{ex}} = 365 \text{ nm}$). (b) Fluorescence emission spectra of NUS-24 bulk powder ($c = 0.2 \text{ mg/mL}$) upon titration with nitrobenzene solution ($1 \times 10^{-3} \text{ M}$) at room temperature ($\lambda_{\text{ex}} = 365 \text{ nm}$). (c) Stern-Völmer plots of NUS-24 nanosheets and bulk powder being titrated with nitrobenzene.

To further understand the difference of quenching behavior between NUS-24 nanosheets and NUS-24 bulk powder, fluorescence titration experiments were performed by gradually adding trace amounts of nitrobenzene into the acetone suspensions containing either NUS-

24 nanosheets or NUS-24 bulk powder (Figure R6). For NUS-24 nanosheets, the measured I_0/I varies linearly with nitrobenzene concentration ($R^2 > 0.99$), and the quenching constant K_{SV} was calculated to be 373740 M^{-1} . In addition, we have noticed that the fluorescence of NUS-24 nanosheets can be quenched faster than that of the bulk powder [K_{SV} constant: 373740 M^{-1} (nanosheets) vs 319973 M^{-1} (bulk)], suggesting a higher sensitivity of nanosheets even under quenching mode. The above discussion and data have been included in the revised manuscript (Supplementary Figure 26 and Supplementary Note 9).

6. Why the nanosheet is particularly selective to Fe³⁺ among all the metal ions tested is not accounted in the manuscript. Further how the Fe³⁺ ion causes the PL quenching requires more clarifications.

Response: We thank the reviewer for this comment. The metal ion induced quenching in this study can be attributed to electron transfer, which is similar to the Fe³⁺ sensing by small organic molecule hexaphenylsilole (HPS) in a previous report (*ChemPhysChem*, 2014, 15, 507–513). To the best of our knowledge, two reasons can be found for the higher sensitivity toward Fe³⁺ in this study. The first reason is the capability of metal ions in accepting transferred electrons. Compared to the alkali metal ions (Li⁺, Na⁺, K⁺), alkaline-earth metal ions (Mg²⁺, Ca²⁺, Ba²⁺, Sr²⁺) and rare earth metal ions (Ce³⁺, Eu³⁺, Tb³⁺, Er³⁺), transition metal ions contain empty and/or half-filled “d” orbitals, which can easily accept electrons. In particular, the configuration of extra-nuclear electron of Fe³⁺ is $1s^2 2s^2 2p^6 3s^2 3p^6 3d^5$, indicating that Fe³⁺ is quite electron-deficient with a strong capability to accept transferred electrons. The second reason is the size of the metal ions. Fe³⁺ has a relatively small size

Dr Dan Zhao

Department of Chemical and Biomolecular Engineering
National University of Singapore
4 Engineering Drive 4, Blk E5, #02-16, Singapore 117585

compared to the rest transition metal ions being tested in this study (Fe^{3+} , 1.1 Å; Ni^{2+} , 1.4 Å; Co^{2+} , 1.5 Å; Cu^{2+} , 1.5 Å; Zn^{2+} , 1.6 Å; Ag^+ , 2.3 Å; Pb^{2+} , 2.3 Å; *ChemPhysChem*, 2014, 15, 507–513), suggesting that Fe^{3+} can interact with NUS-24 more easily triggering more obvious fluorescence quenching than other metal ions. We have included this discussion in the revised manuscript.

Reviewer #2 (Remarks to the Author):

Recommendation: Resubmit after revisions.

Additional Comments:

Dong et al. report tetraphenylethylene (TPE) based π -conjugated 2D porous organic nanosheets named NUS-24 nanosheets for sensing VOCs and Fe³⁺. The materials have been well characterized and the sensing mechanisms have been deeply investigated. The research included in the manuscript could be of interest. However, this manuscript is too primary to publish. The following needed to be addressed before being matured for publication:

Response: We thank the reviewer for the positive comment of our work.

1. What's the advantage of NUS-24 nanosheets as compare with some small organic molecules based fluorescent sensor in sensing Fe³⁺. Indeed, the selectivity and sensitivity of NUS-24 is not that good.

Response: We thank the reviewer for this comment. We agree with the reviewer that the selectivity and sensitivity of NUS-24 for the chemical sensing of Fe³⁺ is not as good as some small organic molecules, especially those containing special chelating sites for Fe³⁺. However, we would like to emphasize that NUS-24 is a pure carbon-based 2D layered polymer without any special chelating site for Fe³⁺. Instead of competing with start-of-the-art molecular

chemical sensors for Fe^{3+} , the aim of this study is to demonstrate better chemical sensing performance of ultrathin 2D nanosheets containing free TPE rotors compared to the monomers and unexfoliated bulk powder.

In order to demonstrate the above point, we chose four small molecules [two of them are monomers TPE-1 and TPE-2, the other two are tetraphenylethylene (TPE) and pyrene], NUS-24 bulk powder, and NUS-24 nanosheets for comparison (Figure R7).

Figure R7. The molecular structures of molecules and materials used in control experiment for metal ion sensing. (a) TPE-1. (b) TPE-2. (c) Tetraphenylethylene (TPE). (d) Pyrene. (e) The

schematic structure of NUS-24 bulk powder. (f) The schematic structure of NUS-24 nanosheets.

It can be clearly seen that the fluorescence quenching of NUS-24 nanosheets caused by the addition of Fe^{3+} is much more obvious than that of the rest controls (Figure R8 a-g). The values of quenching constant K_{SV} , which represent the quenching sensitivity, were calculated to be 27214, 10617, 3632, 2296, 2011, and 1659 M^{-1} for NUS-24 nanosheets, NUS-24 bulk powder, TPE, pyrene, TPE-1, and TPE-2, respectively (Figure R8h).

Figure R8. (a-f) Fluorescence emission spectra of TPE-1 ($c = 1.5 \times 10^{-4}$ M; water/acetone, 90:10, v:v; $\lambda_{\text{ex}} = 360$ nm), TPE-2 ($c = 5.0 \times 10^{-5}$; water/acetone, 90:10, v:v; $\lambda_{\text{ex}} = 360$ nm), TPE ($c = 1.5 \times 10^{-4}$ M; water/acetone, 90:10, v:v; $\lambda_{\text{ex}} = 350$ nm), pyrene ($c = 7.5 \times 10^{-3}$, $\lambda_{\text{ex}} = 350$ nm), NUS-24 bulk powder ($c = 0.1$ mg/mL; acetone; $\lambda_{\text{ex}} = 365$ nm), and NUS-24 nanosheets ($c = 0.1$ mg/mL; acetone; $\lambda_{\text{ex}} = 365$ nm) upon titration with Fe^{3+} solution (1×10^{-2} M) at room temperature. (g) Stern-Völmer plots of TPE-1, TPE-2, TPE, pyrene, NUS-24 bulk powder, and NUS-24 nanosheets being titrated with Fe^{3+} . (h) The quenching percentage and K_{SV} constant of TPE-1, TPE-2, TPE, pyrene, NUS-24 bulk powder, and NUS-24 nanosheets being titrated with Fe^{3+} .

For a more comprehensive comparison, we have also tested other three metal ions including Cu^{2+} , Mn^{2+} , and Co^{2+} . Both the quenching percentages and the K_{SV} constants clearly indicate a higher sensitivity of NUS-24 nanosheets than other controls in sensing those metal ions (Figure R9 a-c). Notably, the fluorescence emissions of the four small organic molecules are barely affected by Co^{2+} , while NUS-24 nanosheets exhibit dramatic quenching in the presence of Co^{2+} (Figure R9c).

Figure R9. (a-c) Quenching percentages and K_{sv} constants of TPE-1, TPE-2, TPE, pyrene, NUS-24 bulk powder, and NUS-24 nanosheets in the chemical sensing of Cu^{2+} , Mn^{2+} , and Co^{2+} . (d) Ion sensing selectivity of TPE-1, TPE-2, TPE, pyrene, NUS-24 bulk powder, and NUS-24 nanosheets.

We have also used $K_{sv}(\text{Fe}^{3+})/K_{sv}(\text{Mn}^{2+})$ and $K_{sv}(\text{Fe}^{3+})/K_{sv}(\text{Cu}^{2+})$ to determine the ion sensing selectivity toward Fe^{3+} . As expected, NUS-24 2D nanosheets have a much higher sensing selectivity toward Fe^{3+} compared to NUS-24 bulk powder and other four small fluorescent molecules (Figure R9d, Table R1), indicating the advantage of nanosheets for metal ion sensing.

Table R1. The values of $K_{sv}(Fe^{3+})/K_{sv}(Cu^{2+})$ and $K_{sv}(Fe^{3+})/K_{sv}(Mn^{2+})$ for TPE-1, TPE-2, TPE, pyrene, NUS-24 bulk powder, and NUS-24 nanosheets.

	$K_{sv}(Fe^{3+})/K_{sv}(Cu^{2+})$	$K_{sv}(Fe^{3+})/K_{sv}(Mn^{2+})$
TPE-1	4.90	1.38
TPE-2	1.38	1.74
TPE	2.81	2.57
Pyrene	2.12	5.28
NUS-24 bulk powder	6.09	11.98
NUS-24 nanosheets	8.39	13.72

All the above results have strongly indicated that NUS-24 nanosheets are more sensitive and selective than their monomers and bulk powder in the chemical sensing of Fe^{3+} . This can be attributed to the large external surface area of the exfoliated 2D nanosheets maximizing the interactions with metal ions. Recently, two reports have demonstrated that the ultrathin 2D MOF nanosheets (*Nature Energy*, 2016, 1, 16184) and 2D layered double hydroxide (LDH) nanosheets (*Angew. Chem., Int. Ed*, 2017, 56, 5867-5871) are better than their 3D bulk state for electrocatalytic oxygen evolution. Similarly, in this study we want to underline that the ultrathin NUS-24 2D nanosheets are better than their monomers and bulk powder for chemical sensing application. The above discussion and data have been included in the revised manuscript (Main text Figure 5 b-d, Supplementary Figures 30-34, 36, and Supplementary Tables 7-8).

2. The authors state that NUS-24 nanosheets can be used in environmental monitoring and biosensing applications of Fe³⁺, I think practical applications data such as imaging Fe³⁺ in living cells etc. should be provided.

Response: We greatly appreciate the reviewer for this valuable comment. According to the reviewer's suggestion, we have tried the biosensing applications of our materials in living HeLa cells. However, NUS-24 nanosheets can barely be taken up by the cells (Figure R10), which unfortunately prevents the further biosensing studies. This result may be attributed to two factors. (1) The NUS-24 nanosheets are pure carbon-based 2D polymer, which is a hydrophobic material based on the contact angle test (98 degree, Figure R11). Therefore, its poor suspension in aqueous solution prevents us from using water as the solvent. Although we have tried using dimethyl sulfoxide (DMSO) as the solvent (other organic solvents can disrupt the cell membrane), cell uptake of NUS-24 nanosheets is still quite difficult. (2) Another reason may be that the NUS-24 nanosheets are too large to be taken up by the cells. Based on the AFM and TEM images, we can clearly see that most NUS-24 nanosheets have a size of 1 ~ 10 micrometers (please see Supplementary Figure 16), which is in the same size range of HeLa cells.

Figure R10. The optical (a) and fluorescence image (b) of the HeLa cells in the presence of NUS-24 nanosheets (0.1 mg mL^{-1}) in DMSO.

Figure R11. The contact angle of NUS-24 bulk powder deposited on a silicon wafer.

We apologize for the overstatement of “biosensing” in the previous version. We have removed such statement in the revised manuscript. Although the application of NUS-24 in biosensing has not been successfully demonstrated in this study, we really appreciate the reviewer for this excellent suggestion, which will surely be explored in our further studies.

3. AIE characteristics (in mixture of acetone and water) of TPE-1, TPE-2, NUS-24 bulk and NUS-24 nanosheets should be investigated and discussed.

Response: We greatly appreciate the reviewer for this valuable comment. We are extremely sorry for the lack of AIE characteristics of TPE-1, TPE-2, NUS-24 bulk powder, and NUS-24 nanosheets in the previous version. This time, we have thoroughly investigated their emission behaviors in acetone/water mixtures with different water fraction (f_w) (Figure R12). As expected, TPE-1 and TPE-2 linkers exhibit typical AIE characteristics. The highest fluorescence intensities were obtained at f_w of 90%, and are 1386- and 2607-fold higher than that in acetone solutions for TPE-1 and TPE-2, respectively. The AIE characteristics of NUS-24 bulk powder and nanosheets are not as obvious as that of TPE-1 and TPE-2, which is mainly because of the restriction of TPE rotors in the polymeric networks. However, we can still observe gradual fluorescence enhancement with the increase of f_w . Interestingly, we found that the fluorescence enhancement of NUS-24 nanosheets became faster at $f_w > 40%$, which can be attributed to the restriction of TPE rotors by re-stacking of nanosheets in solutions with higher f_w . At f_w of 90%, the fluorescence emissions of NUS-24 nanosheets and bulk powder are 1.87- and 1.26-fold higher than that in pure acetone solutions, respectively. In addition, the emission colors of TPE-1, TPE-2, NUS-24 bulk powder and NUS-24 nanosheets differ from each other, which can be clearly seen from the Commission Internationale de L'Eclairage (CIE) chromaticity diagram (Figure R13).

Figure 12. (a, c) Fluorescent spectra of TPE-1 ($c = 1.5 \times 10^{-4}$ M, $\lambda_{\text{ex}} = 360$ nm) and TPE-2 ($c = 5.0 \times 10^{-5}$, $\lambda_{\text{ex}} = 360$ nm) in acetone and acetone/water mixtures. (b, d) Plots of relative emission intensity versus water fraction in acetone/water mixtures of TPE-1 and TPE-2

(Inset: fluorescent photographs with different water fraction). (e, g) Fluorescent spectra of NUS-24 bulk powder ($c = 0.2 \text{ mg mL}^{-1}$, $\lambda_{\text{ex}} = 365 \text{ nm}$) and NUS-24 nanosheets ($c = 20 \mu\text{g mL}^{-1}$, $\lambda_{\text{ex}} = 365 \text{ nm}$) in acetone and acetone/water mixtures. (f, h) Plots of relative emission intensity versus water fraction in acetone/water mixtures of NUS-24 bulk powder and NUS-24 nanosheets.

Figure R13. The CIE chromaticity coordinates of TPE-1 (water/acetone, 90:10, v:v), TPE-2 (water/acetone, 90:10, v:v), NUS-24 bulk powder (acetone), and NUS-24 nanosheets (acetone).

In summary, we have demonstrated the AIE characteristics of TPE-1, TPE-2, NUS-24 bulk powder, and NUS-24 nanosheets. Compared to TPE-1 and TPE-2, the AIE characteristics are not so obvious in NUS-24 bulk powder and nanosheets, which is because of the restricted TPE rotors in the polymeric networks. However, NUS-24 nanosheets exhibit a more obvious AIE behavior in the acetone/water mixtures with water content > 40 vol%, underlining the important role played by the flexible TPE rotors in the AIE characteristics. In addition, we have also demonstrated this point by size-selective VOC sensing (main text Figure 4e). Our finding is similar to a recent study claiming that flexible polymer containing large free volumes of TPE unit is more sensitivity than rigid structure for AIE characteristics (*J. Am. Chem. Soc.* 2017, 139, 5437–5443). The above discussion and data have been included in the revised manuscript (Main text Figure 4a-b, Supplementary Figures 7, 20).

Reviewer #3 (Remarks to the Author):

Zhou and coworkers present a detailed study of an interesting rhombic COF formed from planar 4-connected and 2-connected tetraphenylethylene (TPE) units. The COF is formed as a amorphous layered material due to the irreversible coupling reaction used, however, the authors are able to exfoliate the bulk powder to few-layer thick nanosheets. Screening studies show that the resultant nanosheets exhibit highly sensitive and selective sensing for VOCs and Fe³⁺.

Response: We thank the reviewer for the positive comment of our work.

I have some concerns about the modelling used. Page 9, line 168 states that five-layered nano sheets were formed based on the optimised structure and refers to Fig 1d. Fig 1d, however, is simply a schematic. Searching through the supplementary information for the structure, leads me to Fig S1 and Table S2. Firstly, these two pieces of data are not independent and should be presented together, also the cif file should be included in the SI. More importantly, I am not sure how the simulated parameters were arrived at. In particular, it seems to be that the c lattice parameter might have been chosen to agree with the AFM data. A true structural optimisation would stack layers together as tightly as possible - which is clearly not done here. I understand the difficulty of simulating solvent between layers, and do not ask the authors to do this, however, they do need to be upfront about the constraints applied to their simulation.

Response: We greatly appreciate the reviewer for this valuable comment. According to the reviewer's suggestion, the modelling structures of AA stacking and AB stacking of NUS-24 were first optimized by Materials Studio with the universal force field (UFF) and the QEq method. After that, the density-functional tight-binding (DFTB+) method incorporating dispersion interactions was then used to further refine the obtained structures. DFTB+ is an approximate DFT method based on the second-order expansion of the Kohn–Sham total energy with respect to charge density fluctuations, which is also used for the optimization of COF structures (*Nature Chemistry*, 2015, 7, 905–912). The Slater–Koster library was set as CH, which includes the parameters for the elements of C and H. The convergence criterion of the self-consistent charge (SCC) parameters for electronic minimization was 10^{-4} Ha. Smearing technique was used to achieve the self-consistent field convergence with a smearing value of 0.01 Ha. The lattice dimensions and shapes were optimized simultaneously. We have included the DFTB+ method of structure optimization in Supplementary Note 5.

Following from this point, I would like to see modelling of the bulk NUS-24. Gross parameters such as density and indeed the pore size distribution could be matched and would yield information about the structural change (primarily freeing up the TPE rotors due to the increase in the c lattice constant) occurring upon exfoliation. When modelling the bulk, as layered MOFs/COFs exhibit slipped stacking due to interlayer repulsion and dispersion, I would expect the pore size to decrease somewhat, agreeing better with the 14Å given by the bulk PSD.

Response: We greatly appreciate the reviewer for the valuable comment. The optimized modelling structures of AA stacking and AB stacking of NUS-24 have been obtained by DFTB+ method. As shown in Figure R14 and Table R2, the distance of the adjacent layer in AA stacking is 8.2 Å, which is slightly larger than the previous version without optimization (7.2 Å). The adjacent layers are held together mainly through π - π stacking of TPE phenyl rings. The distance of the adjacent TPE phenyl ring between two layers is about 3.7 Å (Figure R14e), which is also larger than the previous value without optimization (3.3 Å). As the reviewer predicted, this is due to the interlayer repulsion and dispersion.

Figure R14. (a) The simulated crystal structure of NUS-24-AA (along c axis). (b) View of the eclipsed AA stacking structure. (c) Side view (along b axis) of the AA stacking structure. (d)

The dangling TPE rotors in one unit cell represented by phenyl rings with yellow color. (e)

The distance of two adjacent layers and TPE rotors.

Table R2. Parameters of the simulated eclipsed AA stacking structure of NUS-24 optimized by DFTB+.

Structure parameter	NUS-24-AA
Formula	C ₁₄₁ H ₁₀₄
M (g mol ⁻¹)	1798.85
Crystal system	orthorhombic
Space group	P b a n (50)
a / Å	29.4990
b / Å	44.9898
c / Å	8.2402
α / deg.	90
β / deg.	90
γ / deg.	90
V / Å ³	10936.02
Calculated density (g cm ⁻³)	0.2731
surface area (Å ²)	1781.69
Free Volume (Å ³)	8642.77
Occupied Volume (Å ³)	2293.25
Total energy (kcal mol ⁻¹)	540.2

Figure R15. (a) The simulated crystal structure of NUS-24-AB (along b axis). (b) Side view (along a axis) of the AB stacking structure. (c) View of the staggered AB stacking structure. (d) The dangling TPE rotors in one unit cell represented by phenyl rings with red color. (e) The distance of two adjacent layers and TPE rotors.

Table R3. Parameters of the simulated staggered AB stacking structure of NUS-24 optimized by DFTB+.

Structure parameter	NUS-24-AB
Formula	$C_{282}H_{208}$
M (g mol^{-1})	3597.71
Crystal system	orthorhombic
Space group	$P n n a$ (52)
$a / \text{\AA}$	21.8359
$b / \text{\AA}$	18.5797
$c / \text{\AA}$	48.5477
$\alpha / \text{deg.}$	90

β / deg.	90
γ / deg.	90
V / \AA^3	19696.02
Calculated density (g cm^{-3})	0.3033
surface area (\AA^2)	3847.16
Free Volume (\AA^3)	15027.15
Occupied Volume (\AA^3)	4668.87
Total energy (kcal mol^{-1})	1831.5

The AB stacking model of NUS-24 shows a twisted structure (Figure R15b), which is triggered by interlayer repulsion of TPE phenyl rings (Figure R15d). The adjacent layers are held together mainly through π - π stacking interaction of adjacent TPE phenyl rings (Figure R15e), which are about 3.5 \AA apart between two layers. Although the pore size of AA stacking model of NUS-24 (20.2 \AA \times 36.9 \AA) is much larger than that of the AB stacking (9.3 \AA \times 14.0 \AA), the total energy of AA stacking model (540.2 kcal mol^{-1}) is much lower than that of the AB stacking one (1831.5 kcal mol^{-1}). The total energy is calculated using Forcite module by Materials Studio). It is noted that the pore size distribution of NUS-24 bulk powder indicates both micropores (\sim 14 \AA) and mesopores (\sim 28 \AA). However, after exfoliation, NUS-24 nanosheets only contain mesopores (\sim 28 \AA , please see Supplementary Figure 10). Based on the above result, the synthesized NUS-24 bulk powder may adopt an AA stacking model. We agree with the reviewer that layered NUS-24 may exhibit slipped stacking due to interlayer repulsion and dispersion. However, since NUS-24 bulk powder is amorphous in nature, there is no further evidence available to precisely determine the extent of slipped stacking. Having said those, we still thank this valuable suggestion, which can help us to understand the stacking behavior of TPE rotors in the bulk state based on the ideal modelling structure.

The above discussion and data have been included in the revised manuscript (Supplementary Figures 14-15, Supplementary Tables 1-4).

On line 158, the authors state that their modelled structure is “very identical” to their previously published NUS-1. (Side note: it’s either identical or it isn’t). Are the authors sure about this? NUS-1 has a very interesting Kagome structure, which I see no evidence for in the present manuscript. I would think forming the NUS-1 structure would be quite difficult with the two given organic units. This needs to be cleared up.

Response: We thank the reviewer for this comment. We are sorry for the inappropriate description of the modelled structure of NUS-24 in the previous manuscript. We fully agree with the reviewer that NUS-1 is a Kagome structure and is different from the modelled structure of NUS-24 in terms of space group and connection type. However, one thing they have in common is the two-dimensional layered structure, in which adjacent layers are held together by π - π stacking interaction between TPE phenyl rings. In order to avoid the ambiguity, we have removed this sentence “This optimized structure is very identical to our previously reported crystalline 2D TPE-based MOF (NUS-1)” in the revised manuscript.

Given the importance of the freed rotation of the 2-connected TPE units to the proposed mechanism of the framework, this point should also be supported by modelling. It would be quite straightforward for the authors to take a cutout of their 2-connected TPE linker and adjoining 4-connected units and rotate the two free phenyl groups, creating a 1D Potential Energy Surface for the rotation. Repeating this with implicit solvent and

presumably seeing different barriers to rotation would support the role of the rotors in the sensing.

Response: We greatly appreciate the reviewer for the valuable comment. We strongly agree with this crucial suggestion, which is an important piece of information for any molecular rotors. To examine the role of free phenyl groups of TPE-1 linker, we constructed 37 cluster models for NUS-24-AA containing one TPE-1 linker and two TPE-2 linkers (Figure R16). The dangling bonds in the cluster models were saturated by H atoms. The rotor angle around the C–C bonds between two free phenyl groups varied from 0° to 360° with an interval of 10°. The constraint optimization was carried out using the B3LYP functional with 6-31G(d) basis set, followed by the simple point energy calculation at M062X/6-31+G(d) level in a solvent (acetone). The solvent was represented by a polarizable continuum model and the thermal correction to Gibbs free energy was estimated at 298 K. All the DFT calculations were performed by Gaussian 09. We have included the method of energy barrier calculations in the main text (Methods part).

Figure R16. The cluster models of AA stacking model of NUS-24 with different TPE rotor angles. (a) 0°, (b) 50°, (c) 90°, (d) 180°, (e) 240°, (f) 290°. (g) The potential energy as a function of TPE rotor angle between two free phenyl groups.

Figure R16 illustrates the potential energy as a function of rotor angle. Several minima are observed at 50°, 130°, 240° and 320°, implying the existence of stable conformations. While at 90°, 180° and 290°, maxima are seen with relatively higher energy at less favorable

conformations. Three rotation barriers exist with the highest barrier of $14.0 \text{ kcal mol}^{-1}$ at 180° ; and the other two barriers are 5.1 and $8.0 \text{ kcal mol}^{-1}$ at 90° and 290° , respectively. The above discussion and data have been included in the revised manuscript (Main text Figure 1c, Supplementary Figure 3).

Finally, Fig S13 states that the sizes of the VOC molecules were estimated after optimisation in Materials Studio (without mention of method), however, in the main text, DFT calculations were undertaken on all these molecules. The DFT calculations should be used in preference and the details of the estimation provided (e.g. using the total density? vdW radius? In addition, if “square” dimensions are provided, the axes should be marked in the figure (e.g. from where to where corresponds to x Ang measurement?).

Response: We greatly appreciate the reviewer for the valuable comment. According to this suggestion, DFT calculations were performed on the thirteen guest molecules including 1,4-isopropylbenzene, acetone, benzene, dichloromethane, chloroform, cyclohexane, *m*-xylene, *n*-hexane, nitrobenzene, *o*-xylene, *p*-xylene, tetrahydrofuran, and toluene. All the guest molecules were optimized using the B3LYP hybrid functional with 6-31G (d) basis set. We have measured the molecular size of these VOCs by “square” dimensions (Figure R17). Such data have been included in the revised manuscript (Main text Figure 4e and Supplementary Figure 21).

Figure R17. The chemical structure and molecular size of VOCs used in this study. All the guest molecules were optimized using the B3LYP hybrid functional with 6-31G (d) basis set.

We hope our above responses and revisions can clear the reviewers' concerns. Thank you very much for your consideration of this manuscript.

Sincerely,

REVIEWERS' COMMENTS:

Reviewer #1 (Remarks to the Author):

Over all I am satisfied with the changes made and this m/s could be now accepted in Nature Comm.

Reviewer #2 (Remarks to the Author):

The comments have been satisfactorily addressed, I recommend acceptance in the current form

Reviewer #3 (Remarks to the Author):

I thank the authors for their detailed and careful response to my comments and I have no further queries.